# INTRINSIC RIEMANNIAN CLASSIFIERS ON DEFORMED SPD MANIFOLDS: A GENERAL FRAMEWORK

## ABSTRACT

Geometric deep learning, which extends deep learning techniques to non-Euclidean spaces, has gained significant attention in machine learning. Researchers started exploring intrinsic classifiers based on Riemannian geometry to better classify non-Euclidean features in geometric deep learning. However, existing approaches suffer from limited applicability due to their strong reliance on specific geometric properties. This paper proposes a general framework to design intrinsic Riemannian classifiers. Our framework exhibits broad applicability while requiring only minimal geometric properties, enabling its use with a wide range of Riemannian metrics on various Riemannian manifolds. Specifically, we focus on symmetric positive definite (SPD) manifolds and systematically study five families of *deformed* parameterized Riemannian metrics, developing diverse SPD classifiers respecting these metrics. The versatility and effectiveness of the proposed framework are showcased in three applications: radar recognition, human action recognition, and electroencephalography (EEG) classification.

## 1 INTRODUCTION

In recent years, significant advancements have been achieved in deep neural networks (DNNs), enabling them to effectively analyze complex patterns from various types of data, including images, videos, and speech (Hochreiter & Schmidhuber, 1997; Krizhevsky et al., 2012; He et al., 2016; Vaswani et al., 2017). However, most existing models have primarily assumed the underlying data with an Euclidean structure. Recently, a growing body of research has emerged, recognizing that the latent spaces of many applications exhibit non-Euclidean geometries, such as Riemannian geometries (Bronstein et al., 2017). There has been an increasing interest in developing deep learning models tailored for non-Euclidean data, commonly referred to as *Geometric deep learning*. Various frequently-encountered manifolds in machine learning have posed interesting challenges and opportunities, including Lie groups, symmetric positive definite (SPD), Gaussian, spherical, and hyperbolic manifolds (Cavazza et al., 2016; Huang et al., 2017; Vemulapalli et al., 2014; Huang & Van Gool, 2017; Brooks et al., 2019; Ganea et al., 2018; Chen et al., 2021; López et al., 2021; Chen et al., 2023b). These manifolds share an important Riemannian property—their Riemannian operators, such as geodesic, exponential and logarithmic maps, and parallel transportation, often possess closed-form expressions. Leveraging these Riemannian operators, researchers have successfully generalized different types of DNNs into manifolds, which we dub Riemannian networks.

Although Riemannian networks demonstrated success in numerous applications, many approaches still rely on Euclidean spaces for classification, such as tangent spaces (Huang & Van Gool, 2017; Huang et al., 2017; Brooks et al., 2019; Nguyen, 2021; Wang et al., 2021; Nguyen, 2022a;b; Kobler et al., 2022; Wang et al., 2022; Chen et al., 2023c), ambient Euclidean spaces (Wang et al., 2020; Song et al., 2021; 2022), or coordinate systems (Chakraborty et al., 2018). However, these strategies distort the intrinsic geometry of the manifold, undermining the effectiveness of Riemannian networks. Recently, researchers have started developing intrinsic classifiers based on Riemannian geometry for Riemannian networks. Inspired by the idea of hyperplane margin (Lebanon & Lafferty, 2004), Ganea et al. (2018) developed a hyperbolic multinomial logistic regression (MLR) in the Poincaré ball for hyperbolic neural networks (HNNs). Motivated by HNNs, Nguyen & Yang (2023) developed three kinds of SPD MLRs for SPD manifolds based on distinct gyro-structures of SPD manifolds. In parallel, Chen et al. (2023a) proposed a framework for Riemannian classifiers covering the family of Riemannian metrics pulled back from the Euclidean space. However, these classifiers often rely on specific Riemannian properties, limiting their generalizability to other

Riemannian geometries. For instance, the hyperbolic MLR in Ganea et al. (2018) relies on the generalized law of sine, while the SPD MLRs in Nguyen & Yang (2023) rely on the gyro-structures.

This paper presents a general framework for designing Riemannian classifiers for geometric deep learning. In contrast to previous works, our framework only requires the existence of Riemannian logarithm, which is the minimal requirement in extending the Euclidean MLR into manifolds. Since this property is satisfied by common manifolds in machine learning, our framework can be broadly applied to various types of manifolds. More specifically, we mainly focus on SPD manifolds throughout this paper. We leverage the concept of *deformed metrics* to generalize five kinds of popular metrics on SPD manifolds in a unified and systematic manner. Grounded on the proposed Riemannian MLR framework, we propose SPD classifiers induced by these deformed metrics. We further show that some previous works (Nguyen & Yang, 2023; Chen et al., 2023a) are special cases of our framework. Extensive experiments conducted on widely-used SPD benchmarks demonstrate that our proposed SPD classifiers achieve consistent performance gains, outperforming the previous classifiers by about **10%** on human action recognition, and by **4.46%** on electroencephalography (EEG) inter-subject classification. In summary, our main theoretical contributions are as follows: **(a)** We develop a general framework for designing intrinsic classifiers for Riemannian networks. **(b)** On SPD manifolds, we systematically investigate comprehensive generalizations of existing popular Riemannian metrics. **(c)** Based on the derived deformed Riemannian metrics, we propose five families of deformed SPD classifiers.

**Paper structure:** Sec. 2 gives a preliminary review of the geometry of SPD manifolds. Sec. 3 revisits the existing Riemannian MLRs and points out their limitations, then proposes our general framework of Riemannian MLR (Thm. 3.4). Sec. 4 focuses on SPD manifolds, by systematically studying five families of deformed Riemannian metrics (Tab. 1), and proposing five families of SPD classifiers induced by these metrics (Tab. 2). We present the experimental results and some in-depth analysis in Sec. 5. Finally, Sec. 6 summarizes the conclusions.

**Main theoretical results:** We solve the Riemannian margin distance to the hyperplane in Thm. 3.3 and present our Riemannian MLR in Thm. 3.4. For specific SPD manifolds, two existing parametrized metrics can be characterized by the deformation map defined in Def. 4.1. Based on this deformation map, Thm. 4.2 extends and defines three parametrized Riemannian metrics on SPD manifolds in a consistent manner. As these five families of parametrized metrics are pullback metrics, we present Lem. 4.3 for calculating Riemannian MLR (Eq. (14)) under pullback metrics. By this lemma, SPD MLRs under five families of parametrized metrics can be readily obtained, as shown in Tab. 2. Due to page limits, we put all the proofs in the appendix.

## 2 PRELIMINARIES

This section provides a brief review of the basic geometries of SPD manifolds. More details are exposed in App. B. For better clarity, we also summarize notations in App. B.1.

As every smooth manifold admits a Riemannian structure (Do Carmo & Flaherty Francis, 1992), we may use the terms smooth manifold, manifold, or Riemannian manifold interchangeably in this paper. Let $\mathcal{S}_{++}^n$ be the set of $n \times n$ symmetric positive definite (SPD) matrices. As shown in Arsigny et al. (2005), $\mathcal{S}_{++}^n$ is an open submanifold of the Euclidean space $\mathcal{S}^n$ consisting of symmetric matrices. In machine learning, there are five kinds of popular Riemannian metrics on $\mathcal{S}_{++}^n$: Affine-Invariant Metric (AIM) (Pennec et al., 2006), Log-Euclidean Metric (LEM) (Arsigny et al., 2005), Power-Euclidean Metrics (PEM) (Dryden et al., 2010), Log-Cholesky Metric (LCM) (Lin, 2019), and Bures-Wasserstein Metric (BWM) (Bhatia et al., 2019). Note that, when power equals 1, the PEM is reduced to the Euclidean Metric (EM). Some of these basic metrics have been generalized into parametrized families of metrics. We define $\mathbf{ST} = \{(\alpha, \beta) \in \mathbb{R}^2 \mid \min(\alpha, \alpha + n\beta) > 0\}$, and denote the O($n$)-invariant Euclidean metric on $\mathcal{S}^n$ (Thanwerdas & Pennec, 2023) as

$$\langle V, W \rangle^{(\alpha,\beta)} = \alpha \langle V, W \rangle + \beta \operatorname{tr}(V) \operatorname{tr}(W), \text{ with } (\alpha, \beta) \in \mathbf{ST}. \tag{1}$$

By O($n$)-invariant Euclidean metric on $\mathcal{S}^n$, Thanwerdas & Pennec (2023) generalized AIM, LEM, and EM into two-parameters families of O($n$)-invariant metrics, *i.e.,* $(\alpha, \beta)$-AIM, $(\alpha, \beta)$-LEM, and $(\alpha, \beta)$-EM, with $(\alpha, \beta) \in \mathbf{ST}$. We denote the metric tensor of $(\alpha, \beta)$-AIM, $(\alpha, \beta)$-LEM, $(\alpha, \beta)$-EM, LCM, and BWM as $g^{(\alpha,\beta)\text{-AIM}}$, $g^{(\alpha,\beta)\text{-LEM}}$, $g^{(\alpha,\beta)\text{-EM}}$, $g^{\text{BWM}}$, and $g^{\text{LEM}}$, respectively. We leave their properties and the formula of associated Riemannian operators in App. B.3. Although there

also exist other metrics on SPD manifolds (Thanwerdas & Pennec, 2019b; 2022a; 2023), their lack of closed-form Riemannian operators makes them problematic to be applied in machine learning.

Now, we review the definition of pullback metric, a common technique in Riemannian geometry.

**Definition 2.1** (Pullback Metrics). Suppose $\mathcal{M}, \mathcal{N}$ are smooth manifolds, $g$ is a Riemannian metric on $\mathcal{N}$, and $f : \mathcal{M} \to \mathcal{N}$ is smooth. Then the pullback of $g$ by $f$ is defined point-wisely,

$$(f^*g)_p(V_1, V_2) = g_{f(p)}(f_{*,p}(V_1), f_{*,p}(V_2)), \tag{2}$$

where $p \in \mathcal{M}$, $f_{*,p}(\cdot)$ is the differential map of $f$ at $p$, and $V_i \in T_p\mathcal{M}$. If $f^*g$ is positive definite, it is a Riemannian metric on $\mathcal{M}$, which is called the pullback metric defined by $f$.

## 3 RIEMANNIAN MULTICLASS LOGISTIC REGRESSION

Inspired by Lebanon & Lafferty (2004), Ganea et al. (2018); Nguyen & Yang (2023); Chen et al. (2023a) extended the Euclidean MLR into hyperbolic and SPD manifolds. However, these classifiers were developed in an ad-hoc manner, relying on specific Riemannian properties, such as the generalized law of sines, gyro-structures, and Euclidean pullback metrics, which limits their generality. In this section, we first revisit several existing MLRs and then propose our Riemannian classifiers with minimal geometric requirements.

### 3.1 REVISITING THE EXISTING MLRS

We now summarize the abstract ideas behind the development of Riemannian classifiers by Lebanon & Lafferty (2004); Ganea et al. (2018); Nguyen & Yang (2023); Chen et al. (2023a).

Given $C$ classes, the Euclidean MLR computes the multinomial probability of each class as follows:

$$\forall k \in \{1, \dots, C\}, \quad p(y = k \mid x) \propto \exp\left(\langle a_k, x \rangle - b_k\right), \quad b_k \in \mathbb{R}, x, a_k \in \mathbb{R}^n \backslash \{\mathbf{0}\}. \tag{3}$$

As shown in previous work such as Ganea et al. (2018); Chen et al. (2023a), the Euclidean MLR can be reformulated by the margin distance to the hyperplane:

$$p(y = k \mid x) \propto \exp(\text{sign}(\langle a_k, x - p_k \rangle)\|a_k\| d(x, H_{a_k, p_k})), \quad p_k, x \in \mathbb{R}^n, \text{ and } a_k \in \mathbb{R}^n \backslash \{\mathbf{0}\}, \tag{4}$$

where $\langle a_k, p_k \rangle = b_k$, and the hyperplane $H_{a_k, p_k}$ is defined as:

$$H_{a_k, p_k} = \{x \in \mathbb{R}^n : \langle a_k, x - p_k \rangle = 0\}. \tag{5}$$

It is now natural to adapt Eq. (4) and Eq. (5) to manifolds by Riemannian operators:

$$p(y = k \mid S) \propto \exp(\text{sign}(\langle \tilde{A}_k, \text{Log}_{P_k}(S) \rangle_{P_k})\|\tilde{A}_k\|_{P_k} \tilde{d}(S, \tilde{H}_{\tilde{A}_k, P_k})), \tag{6}$$

$$\tilde{H}_{\tilde{A}_k, P_k} = \{S \in \mathcal{M} : g_{P_k}(\text{Log}_{P_k} S, \tilde{A}_k) = \langle \text{Log}_{P_k} S, \tilde{A}_k \rangle_{P_k} = 0\}, \tag{7}$$

where $P_k \in \mathcal{M}$, $\tilde{A}_k \in T_{P_k}\mathcal{M}\backslash\{\mathbf{0}\}$, and $g_{P_k}(\cdot, \cdot)$ $(\text{Log}_{P_k}(\cdot))$ is the Riemannian metric (Riemannian logarithmic map) at $P_k$. The margin distance is defined as an infimum:

$$\tilde{d}(S, \tilde{H}_{\tilde{A}_k, P_k})) = \inf_{Q \in \tilde{H}_{\tilde{A}_k, P_k}} d(S, Q). \tag{8}$$

The MLRs proposed in Lebanon & Lafferty (2004); Ganea et al. (2018); Nguyen & Yang (2023); Chen et al. (2023a) can be viewed as different implementations of Eq. (6)-Eq. (8).

To calculate the MLR in Eq. (6), one has to compute the associated Riemannian metrics, logarithmic maps, and margin distance. The associated Riemannian metrics and logarithmic maps often have closed-form expressions in machine learning applications. However, the computation of the margin distance can be challenging. On the Poincaré ball of hyperbolic manifolds, the generalized law of sines simplifies the calculation of Eq. (8) (Ganea et al., 2018). However, the generalized law of sines is not universally guaranteed on other manifolds, even when considering other metrics on hyperbolic spaces. Additionally, Chen et al. (2023a) demonstrated that when working with metrics pulled back from Euclidean spaces, closed-form solutions of margin distance can be readily obtained. For curved manifolds, solving Eq. (8) would become a non-convex optimization problem. To address this challenge, Nguyen & Yang (2023) defined gyro-structures on SPD manifolds and proposed a pseudo-gyrodistance to calculate the margin distance. It is also important to note that gyro-structures do not generally exist in other types of Riemannian metrics. *In summary, the aforementioned methods often rely on specific properties of their associated Riemannian metrics, which usually do not generalize to other Riemannian metrics.*

### 3.2 RIEMANNIAN MLR

Recalling Eq. (6) and (7), the weakest requirement of extending Euclidean MLR in manifolds is the well-definedness of $\mathrm{Log}_{P_k}(S)$ for each $k$. If the $\mathrm{Log}_{P_k}(S)$ does not exist, even the hyperplane Eq. (6) and (7) are all ill-defined. In this subsection, we will develop Riemannian MLR depending solely on the existence of the Riemannian logarithm, without any additional requirements, such as gyro structures and generalized lay of sines. In the following, we always assume the existence of the Riemannian logarithm. We start by reformulating the Euclidean margin distance to the hyperplane from another perspective and then present our Riemannian MLR.

*Remark* 3.1. In terms of the existence of the Riemannian logarithm, we make the following remarks.

(a). The existence of $\mathrm{Log}_{P_k}(S)$ is more precisely described as $S$ lies in the local injectivity radius (Groisser, 2004) of $P_k$. However, in machine learning, cases are much easier. The Riemannian logarithm always exists for a pair of data on many manifolds or metrics, such as the five metrics on SPD manifolds mentioned in Sec. 2, different metrics on hyperbolic manifolds (Cannon et al., 1997), spherical manifolds (Chakraborty, 2020), and various types of Lie groups (Iserles et al., 2000). Therefore, without loss of generality, this paper assumes that Riemannian logarithm $\mathrm{Log}_{P_k}(S)$ always exists on the manifold $\mathcal{M}$.

(b). This property is a much weaker condition compared to the existence of the gyro structure. For instance, on SPD manifolds, EM and BWM (Thanwerdas & Pennec, 2023) are incomplete, which might bring problems when defining gyro operations (Nguyen & Yang, 2023, Eqs. (1-2)). In contrast, they have the Riemannian logarithm for any data pair.

As we discussed before, on manifolds, obtaining the margin distance of Eq. (8) could be challenging. Inspired by Nguyen & Yang (2023), we resort to another perspective to reinterpret Euclidean margin distance. In Euclidean space, the margin distance is equivalent to

$$d(x, H_{a,p})) = \sin(\angle xpy^*)d(x, p) \tag{9}$$

where $y^*$ is given by

$$y^* = \underset{y \in H_{a,p} \setminus \{p\}}{\arg\max} \ (\cos \angle xpy) \tag{10}$$

The Riemannian counterparts of Euclidean trigonometry and distance in Eq. (9) and Eq. (10) are Riemannian trigonometry and geodesic distance. Therefore, we can readily extend the margin distance to manifolds.

**Definition 3.2** (Riemannian Margin Distance). Let $\tilde{H}_{\tilde{A},P}$ be a Riemannian hyperplane defined in Eq. (7), and $S \in \mathcal{M}$. The Riemannian margin distance from $S$ to $\tilde{H}_{\tilde{A},P}$ is defined as

$$d(S, \tilde{H}_{\tilde{A},P}) = \sin(\angle SPQ^*)d(S, P), \tag{11}$$

where $d(S, P)$ is the geodesic distance, and $Q^* = \underset{Q \in \tilde{H}_{\tilde{A},P} \setminus \{P\}}{\arg\max} \ (\cos \angle SPQ)$. The initial velocities of geodesics define $\cos \angle SPQ$:

$$\cos \angle SPQ = \frac{\langle \mathrm{Log}_P Q, \mathrm{Log}_P S \rangle_P}{\| \mathrm{Log}_P Q \|_P, \| \mathrm{Log}_P S \|_P}, \tag{12}$$

where $\langle \cdot, \cdot \rangle_P$ is the Riemannian metric at $P$, and $\| \cdot \|_P$ is the associated norm.

The following theorem can calculate the Riemannian margin distance.

**Theorem 3.3.** *The Riemannian margin distance defined in Eq.* (11) *is given as*

$$d(S, \tilde{H}_{\tilde{A},P}) = \frac{|\langle \mathrm{Log}_P S, \tilde{A} \rangle_P|}{\|\tilde{A}\|_P}. \tag{13}$$

Putting the margin distance (Eq. (13)) into Eq. (6), RMLR can be obtained. However, In Eq. (6), as $P_k$ varies during training, $\tilde{A}_k$ becomes a non-Euclidean parameter. Inspired by Chen et al. (2023a), $\tilde{A}_k$ can be generated by parallel transportation, or if $\mathcal{M}$ admits Lie group structures, Lie group translation (we focus on left translation). Now, we give the final expression of our Riemannian MLR.

**Theorem 3.4** (Riemannian MLR). *Given a Riemannian manifold $\{\mathcal{M}, g\}$, the Riemannian MLR induced by g is*

$$p(y = k \mid S \in \mathcal{M}) \propto \exp(\langle \mathrm{Log}_{P_k} S, \tilde{A}_k \rangle_{P_k}), \tag{14}$$

*where $P_k \in \mathcal{M}$, $\tilde{A}_k \in T_{P_k}\mathcal{M}\backslash\{\mathbf{0}\}$, and $\mathrm{Log}$ is the Riemannian logarithm. $\tilde{A}_k$ can be generated by parallel transportation, or, if $\mathcal{M}$ admits a Lie group operation $\odot$, Lie group left translation:*

$$\tilde{A}_k = \Gamma_{Q \to P_k} A_k, \tag{15}$$

$$\tilde{A}_k = L_{P_k \odot Q_\odot^{-1} *, Q} A_k, \tag{16}$$

*where $Q \in \mathcal{M}$ is a fixed point, $A_k \in T_Q\mathcal{M}\backslash\{0\}$, $\Gamma$ is the parallel transportation along geodesic connecting $Q$ and $P_k$, and $L_{P_k \odot Q_\odot^{-1} *, Q}$ denotes the differential map at $Q$ of left translation $L_{P_k \odot Q_\odot^{-1}}$ with $P_k \odot Q_\odot^{-1}$ denoting Lie group product and inverse. We call Eq. (14) + Eq. (15) the Riemannian MLR by parallel transportation, and Eq. (14) + Eq. (16) the Riemannian MLR by left translation.*

*Remark* 3.5. Compared with the specific classifiers on hyperbolic or SPD manifolds in Ganea et al. (2018); Nguyen & Yang (2023); Chen et al. (2023a), our framework enjoys broader applicability, as our framework only requires the existence of Riemannian logarithm, which is commonly satisfied by most manifolds encountered in machine learning. Besides, optimization of $\tilde{A}_k$ in our framework incorporates the methods in Ganea et al. (2018); Nguyen & Yang (2023); Chen et al. (2023a). Interestingly, our definition of Riemannian MLR is a natural generalization of Euclidean MLR: When $\mathcal{M} = \mathbb{R}^n$, our Riemannian MLR becomes the Euclidean MLR. Please refer to App. C for more details.

## 4 SPD MLRs

In this section, we aim to propose SPD MLRs based on our previously defined Riemannian MLR framework. To achieve this, we first systematically discuss the diverse geometries of SPD manifolds. Subsequently, we will develop five families of SPD MLRs.

### 4.1 Five families of Deformed Geometries of SPD manifolds

As discussed in Sec. 2, there are five kinds of popular Riemannian metrics on SPD manifolds. In Thanwerdas & Pennec (2019a), $(\alpha, \beta)$-AIM is further generalized into three-parameters families of metrics by the pullback of matrix power function $\mathrm{P}_\theta$ and scaled by $\frac{1}{\theta^2}$, *i.e.*, $(\theta, \alpha, \beta)$-AIM. Thanwerdas & Pennec (2022a) identified the alpha-Procrustes metric (Minh, 2022) with one-parameter families of BWM pulled back by $\mathrm{P}_{2\theta}$ and scaled by $\frac{1}{4\theta^2}$, which we call as $2\theta$-BWM in this paper. Besides, the pullback of the power function can be viewed as deformation. The family of $(\theta, \alpha, \beta)$-AIM comprises $(\alpha, \beta)$-AIM for $\theta = 1$ and includes $(\alpha, \beta)$-LEM with $\theta \to 0$ (Thanwerdas & Pennec, 2019a). Similarly, $2\theta$-BWM becomes BWM with $\theta = 0.5$ (Thanwerdas & Pennec, 2022a).

Inspired by the deforming utility of power function, in the following, we generalize the basic LCM, LEM, and EM into parameterized families of metrics, and systematically study the deformation between different families of metrics. We first define a deformation map between SPD manifolds to present a unified discussion encompassing all metrics.

**Definition 4.1** (Power Deformation Map). A power deformation map $\phi_\theta : \mathcal{S}_{++}^n \to \mathcal{S}_{++}^n$ is defined as $\phi_\theta(S) = \frac{1}{|\theta|}S^\theta$, where $|\cdot|$ denotes the absolute value and $\theta \in \mathbb{R}\backslash\{0\}$.

Easy computation shows that $2\theta$-BWM is the pullback metrics by $\phi_{2\theta}$ from the standard BWM. So is $(\theta, \alpha, \beta)$-AIM from the $(\alpha, \beta)$-AIM. Inspired by this observation, we use the power deformation map to consistently generalize the standard EM, LCM, and LEM.

**Theorem 4.2.** *We define the $(\theta, \alpha, \beta)$-LEM, $(\theta, \alpha, \beta)$-EM, and $\theta$-LCM as the pullback metrics by $\phi_\theta$ from the $(\alpha, \beta)$-LEM, $(\alpha, \beta)$-EM, and LCM, respectively, with $(\alpha, \beta) \in \mathbf{ST}$. Then $(\theta, \alpha, \beta)$-LEM interpolates between $(\alpha, \beta)$-LEM ($\theta \to 0$) and $(\alpha, \beta)$-LEM ($\theta = 1$), while $(\theta, \alpha, \beta)$-EM interpolates between $(\alpha, \beta)$-LEM ($\theta \to 0$) and LCM ($\theta = 1$).*

*Define $\log_{*,P}$ as the differential map at $P \in \mathcal{S}_{++}^n$ of matrix logarithm, $\tilde{g}(V_1, V_2) = \frac{1}{2}\langle V_1, V_2 \rangle - \frac{1}{4}\langle \mathbb{D}(V_1), \mathbb{D}(V_2) \rangle$, where $\mathbb{D}(V_i)$ is a diagonal matrix consisting of the diagonal elements of $V_i$. Then $\theta$-LCM interpolates between $\tilde{g}$-LEM ($\theta = 0$) and itself ($\theta = 1$), with $\tilde{g}$-LEM defined as*

$$\langle V, W \rangle_P = \tilde{g}(\log_{*,P}(V), \log_{*,P}(W)), \forall P \in \mathcal{S}_{++}^n, \forall V, W \in T_P \mathcal{S}_{++}^n. \tag{17}$$

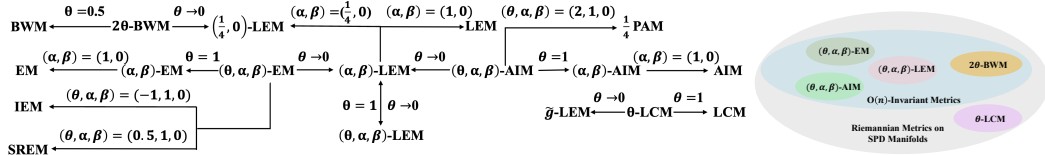

Figure 1: Illustration on the deformation (**left**) and Venn diagram (**right**) of metrics on SPD manifolds, where IEM, SREM, and $\frac{1}{4}$ PAM denotes Inverse Euclidean Metric, Square Root Euclidean Metric, and Polar Affine Metric scaled by $1/4$.

So far, all five standard popular Riemannian metrics on SPD manifolds have been generalized into parameterized families of metrics. We summarize their associated properties in Tab. 1 and present their theoretical relation in Fig. 1. We leave technical details in Apps. D.1 and D.2.

Table 1: Properties of five families of parameterized Riemannian metrics on SPD manifolds.

| Name | Riemannian Metric | Properties |
|---|---|---|
| $(\theta, \alpha, \beta)$-LEM | $\phi_\theta^* g^{(\alpha,\beta)\text{-LEM}}$ | Lie Group Bi-Invariance, O($n$)-Invariance, Geodesically Completeness |
| $(\theta, \alpha, \beta)$-AIM | $\phi_\theta^* g^{(\alpha,\beta)\text{-AIM}}$ | Lie Group Left-Invariance, O($n$)-Invariance, Geodesically Completeness |
| $(\theta, \alpha, \beta)$-EM | $\phi_\theta^* g^{(\alpha,\beta)\text{-EM}}$ | O($n$)-Invariance |
| $\theta$-LCM | $\phi_\theta^* g^{\text{LCM}}$ | Lie Group Bi-Invariance, Geodesically Completeness |
| $2\theta$-BWM | $\phi_{2\theta}^* g^{\text{BWM}}$ | O($n$)-Invariance |

## 4.2 FIVE FAMILIES OF SPD MLRS

This section presents five families of specific SPD MLRs by leveraging our general framework in Thm. 3.4 and metrics discussed in Sec. 4.1. We focus on generating $\tilde{A}$ by parallel transportation from the identity matrix, except for $2\theta$-BWM. Since the parallel transportation under $2\theta$-BWM would undermine numerical stability (please refer to App. E.2.1 for more details.), we rely on left translation to generate $\tilde{A}$ for the MLRs under $2\theta$-BWM.

As the five families of metrics presented in Sec. 4.1 are pullback metrics, we first present a general result regarding Riemannian MLRs under pullback metrics.

**Lemma 4.3** (Riemannian MLRs Under Pullback Metrics). *Supposing $\{\mathcal{N}, g\}$ is a Riemannian manifold and $\phi : \mathcal{M} \to \mathcal{N}$ is a diffeomorphism between manifolds, the Riemannian MLR by parallel transportation (Eq. (14) + Eq. (15)) on $\mathcal{M}$ under $\tilde{g} = \phi^* g$ can be obtained by $g$:*

$$p(y = k \mid S \in \mathcal{M}) \propto \exp(\tilde{g}_{P_k}(\tilde{\text{Log}}_{P_k} S, \tilde{\Gamma}_{Q \to P_k} A_k)), \tag{18}$$

$$= \exp\left[ g_{\phi(P_k)}(\text{Log}_{\phi(P_k)} \phi(S), \Gamma_{\phi(Q) \to \phi(P_k)} \phi_{*,Q}(A_k)) \right], \tag{19}$$

*where $\tilde{\text{Log}}, \tilde{\Gamma}$ are Riemannian logarithm and parallel transportation under $\tilde{g}$, and $\text{Log}, \Gamma$ are the counterparts under $g$.*

*Furthermore, if $\mathcal{N}$ has a Lie group operation $\odot$, $\mathcal{M}$ could be endowed with a Lie group structure $\tilde{\odot}$ by $f$. The Riemannian MLR by left translation (Eq. (14) + Eq. (16)) on $\mathcal{M}$ under $\tilde{g}$ and $\tilde{\odot}$ can be calculated by $g$ and $\odot$:*

$$p(y = k \mid S \in \mathcal{M}) \propto \exp(\tilde{g}_{P_k}(\tilde{\text{Log}}_{P_k} S, \tilde{L}_{\tilde{R}_k*, Q} A_k)), \tag{20}$$

$$= \exp\left[ g_{\phi(P_k)} \left( \text{Log}_{\phi(P_k)} \phi(S), L_{R_k*, \phi(Q)} \circ \phi_{*,Q}(A_k) \right) \right], \tag{21}$$

*where $\tilde{R}_k = P_k \tilde{\odot} Q_{\tilde{\odot}}^{-1}$, $R_k = \phi(P) \odot \phi(Q)_\odot^{-1}$, and $\tilde{L}_{P_k \tilde{\odot} Q^{-1}_{\tilde{\odot}}}$ is the left translation under $\tilde{\odot}$*

For MLRs on SPD manifolds, we set $Q = I$. For $(\theta, \alpha, \beta)$-LEM, $(\theta, \alpha, \beta)$-AIM, $(\theta, \alpha, \beta)$-EM, and $\theta$-LCM, MLRs can be readily obtained by Lem. 4.3. For $2\theta$-BWM, we resort to a newly developed Lie group operation (Thanwerdas & Pennec, 2022b) defined as $S_1 \odot S_2 = L_1 S_2 L_1^T$ with $L_1 = \text{Chol}(S_1)$ as the Cholesky decomposition. We also propose a numerically stable backpropagation for the Lyapunov operator in $2\theta$-BWM (please refer to App. E.2.2 for technical details.).

By Lem. 4.3, we can readily obtain the specific SPD MLRs induced from five families of parameterized metrics. As $2 \times 2$ SPD matrices can be embedded into $\mathbb{R}^3$ as an open cone (Yair et al., 2019), we also illustrate SPD hyperplanes induced by five families of metrics in Fig. 2. The SPD MLRs

Table 2: Five families of SPD MLRs. For simplicity, we omit the subscripts $k$ of $A_k$ and $P_k$.

| Metrics | $p(y = k \mid S \in \mathcal{S}_{++}^n) =$ | Prototype |
|---|---|---|
| $(\theta, \alpha, \beta)$-LEM | $\exp\left[\theta^2 \langle \log(S) - \log(P), A \rangle^{(\alpha,\beta)}\right]$ | Eq. (19) |
| $(\theta, \alpha, \beta)$-AIM | $\exp\left[\langle \log(P^{-\frac{\theta}{2}} S^\theta P^{-\frac{\theta}{2}}), \theta A \rangle^{(\alpha,\beta)}\right]$ | Eq. (19) |
| $(\theta, \alpha, \beta)$-EM | $\exp\left[\frac{1}{\theta} \langle S^\theta - P^\theta, A \rangle^{(\alpha,\beta)}\right]$ | Eq. (19) |
| $\theta$-LCM | $\exp\left[\mathrm{sgn}(\theta) \langle \lfloor \tilde{K} \rfloor - \lfloor \tilde{L} \rfloor + \sqrt{|\theta|} \left[\mathrm{Dlog}(\mathbb{D}(\tilde{K})) - \mathrm{Dlog}(\mathbb{D}(\tilde{L}))\right], \lfloor A \rfloor + \frac{\sqrt{|\theta|}}{2} \mathbb{D}(A) \rangle\right]$ | Eq. (19) |
| $2\theta$-BWM | $\exp\left[\frac{\mathrm{sgn}(\theta)}{2} \langle (P^{2\theta} S^{2\theta})^{\frac{1}{2}} + (S^{2\theta} P^{2\theta})^{\frac{1}{2}} - 2P^{2\theta}, \mathcal{L}_{P^{2\theta}}(\bar{L} A \bar{L}^\top) \rangle\right]$ | Eq. (21) |

Figure 2: Conceptual illustration of SPD hyperplanes induced by five families of Riemannian metrics. In each subfigure, the black dots are symmetric positive semi-definite matrices, denoting the boundary of $\mathcal{S}_{++}^2$. The blue, red, and yellow dots denote three SPD hyperplanes.

are presented in Tab. 2. We leave technical details in App. E.3. The notations in Tab. 2 are defined in the following. By abuse of notation, we omit the subscripts $k$ of $A_k$ and $P_k$. For $P \in \mathcal{S}_{++}^n$ and $A \in T_I \mathcal{S}_{++}^n \setminus \{0\}$, we make the following notations. We denote $\log(\cdot)$ as the matrix logarithm, $\mathcal{L}_P(V)$ as the solution to the matrix linear system $\mathcal{L}_P[V]P + P\mathcal{L}_P[V] = V$, which is known as the Lyapunov operator, $\mathrm{Dlog}(\cdot)$ as the diagonal element-wise logarithm, $\lfloor \cdot \rfloor$ as the strictly lower part of a square matrix, and $\mathbb{D}(\cdot)$ as a diagonal matrix with diagonal elements of a square matrix. Besides, $\log_{*,P}$ is the differential maps at $P$. We set $\tilde{K} = \mathrm{Chol}(S^\theta)$, $\tilde{L} = \mathrm{Chol}(P^\theta)$, and $\bar{L} = \mathrm{Chol}(P^{2\theta})$.

*Remark* 4.4. Our SPD MLRs extend the SPD MLRs in Nguyen & Yang (2023); Chen et al. (2023a):

Nguyen & Yang (2023) introduced SPD MLRs induced by gyro-structures under standard LEM, LCM, and AIM, and Chen et al. (2023a) discussed SPD MLRs under pullback Euclidean metrics and specifically focused on $(\alpha, \beta)$-LEM. However, our work covers their theoretical results. In detail, Nguyen & Yang (2023, Thms. 2.23-2.25) and Chen et al. (2023a, Thm. 3 and Prop. 7) can be readily obtained by our Thm. 3.3 and Thm. 3.4.

Furthermore, our approach extends the scope of prior work as neither Chen et al. (2023a) nor Nguyen & Yang (2023) explored SPD MLRs based on EM and BWM. The formal definition of gyro-operations in Nguyen & Yang (2023, Eq. (1)) implicitly requires geodesic completeness, whereas EM and BWM are imcomplete. As neither EM nor BWM belongs to pullback Euclidean metrics, the framework presented in Chen et al. (2023a) cannot be applied to these metrics either. To the best of our knowledge, our work is the **first** to apply EM and BWM to establish Riemannian neural networks, opening up new possibilities for utilizing these metrics in machine learning applications.

## 5 EXPERIMENTS

Following the previous work (Huang & Van Gool, 2017; Brooks et al., 2019; Kobler et al., 2022), we evaluate the performance of our classifiers in three different applications: radar recognition on the Radar dataset (Brooks et al., 2019), human action recognition on the HDM05 dataset (Müller et al., 2007), and Brain Computer Interface (BCI) application on the Hinss2021 dataset (Hinss et al., 2021). For the radar recognition and human action recognition tasks, we apply our classifiers to the baseline SPDNet[1] (Huang & Van Gool, 2017). On the BCI application, we utilize our classifiers with the state-of-the-art SPD neural network, SPD domain-specific momentum batch normalization (SPDDSMBN [2]) (Kobler et al., 2022), which is an improved version of SPDNetBN (Brooks et al., 2019). Please refer to App. B.4 for a quick review of these baseline models. We use the standard cross entropy loss as the training objective and optimize the parameters with the Riemannian AMS-Grad optimizer (Bécigneul & Ganea, 2018). We denote the network architecture as $[d_0, d_1, \cdots, d_L]$, where the dimension of the parameter in the $i$-th BiMap layer (App. B.4) is $d_i \times d_{i-1}$. The learning

---

[1] https://proceedings.neurips.cc/paper_files/paper/2019/file/6e69ebbfad976d4637bb4b39de261bf7-Supplemental.zip

[2] https://github.com/rkobler/TSMNet

rate for the Radar and HDM05 datasets is $1e^{-2}$, and the batch size is 30. For the Hinss2021 dataset, following Kobler et al. (2022), the learning rate is $1e^{-3}$ with a $1e^{-4}$ weight decay, and batch size is 50. The maximum training epoch is 200, 200, and 50, respectively.

**Datasets and preprocessing:** **Radar**[3] dataset (Brooks et al., 2019) consists of 3,000 synthetic radar signals. Following the protocol in Brooks et al. (2019), each signal is split into windows of length 20, resulting in 3,000 SPD covariance matrices of $20 \times 20$ equally distributed in 3 classes. **HDM05**[4] dataset (Müller et al., 2007) contains 2,273 skeleton-based motion capture sequences executed by various actors. Each frame consists of 3D coordinates of 31 joints of the subjects, and each sequence can be, therefore, modeled by a $93 \times 93$ covariance matrix. Following the protocol in Brooks et al. (2019), we trim the dataset down to 2086 sequences scattered throughout 117 classes by removing some under-represented classes. **Hinss2021**[5] dataset (Hinss et al., 2021) is a recent competition dataset consisting of EEG signals for mental workload estimation. The dataset is used for two types of experiments: inter-session and inter-subject, which are modeled as domain adaptation problems. Recently, geometry-aware methods have shown promising performance in EEG classification (Yair et al., 2019; Kobler et al., 2022; Abdel-Ghaffar et al., 2022). We choose the SOTA method, SPDDSMBN (Kobler et al., 2022), as our baseline model on this dataset. We follow Kobler et al. (2022) to carry out preprocessing and finally extract $40 \times 40$ SPD covariance matrices (see App. F.2 for more details).

**Implementation Details:** In the baseline models, namely SPDNet and SPDDSMBN, the Euclidean MLR in the co-domain of matrix logarithm (matrix logarithm + FC + softmax) is used for classification. Following the terminology in Chen et al. (2023a), we call this classifier as LogEig MLR. To evaluate the performance of our intrinsic classifiers, we substitute the LogEig MLR in SPDNet and SPDDSMBN with our SPD MLRs. We implement our SPD MLRs induced from five parameterized metrics. In line with the previous work (Brooks et al., 2019; Kobler et al., 2022), we use accuracy as the scoring metric for the Radar and HDM05 datasets, and balanced accuracy (*i.e.,*the average recall across classes) for the Hinss2021 dataset. 10-fold experiments on Radar and HDM05 datasets are carried out with randomized initialization and split, while models are fit and evaluated with a randomized leave 5% of the sessions (inter-session) or subjects (inter-subject) out cross-validation (CV) scheme on the Hinss2021 dataset.

**Hyper-parameters:** We implement the SPD MLRs induced by not only five standard metrics, *i.e.,*LEM, AIM, EM, LCM, and BWM, but also five families of parameterized metrics. Therefore, in our SPD MLRs, we have a maximum of three hyper-parameters, *i.e.,*$\theta, \alpha, \beta$, where $(\alpha, \beta)$ are associated with O($n$)-invariance and $\theta$ controls deformation. For $(\alpha, \beta)$ in $(\theta, \alpha, \beta)$-LEM, $(\theta, \alpha, \beta)$-AIM, and $(\theta, \alpha, \beta)$-EM, recalling Eq. (1), $\alpha$ is a scaling factors, while $\beta$ measures the relative significance of traces. As scaling is less important (Thanwerdas & Pennec, 2019a), we set $\alpha = 1$. As for the value of $\beta$, we select it from a predefined set: $\{1, 1/n, 1/n^2, 0, -1/n + \epsilon, -1/n^2\}$, where $n$ is the dimension of input SPD matrices in SPD MLRs. The purpose of including $\epsilon \in \mathbb{R}_+$ is to ensure O($n$)-invariance ($(\alpha, \beta) \in$ **ST**). These chosen values for $\beta$ allow for amplifying, neutralizing, or suppressing the trace components, depending on the characteristics of the datasets. For the deformation factor $\theta$, we roughly select its value around its deformation boundary, detailed in App. F.1.

Table 3: Accuracy comparison of SPDNet with and without SPD MLRs on the Radar dataset.

| Network Architectures | SPDNet | $(\theta, \alpha, \beta)$-AIM | $(\theta, \alpha, \beta)$-EM | | $(\theta, \alpha, \beta)$-LEM | | $2\theta$-BWM | | $\theta$-LCM | |
|---|---|---|---|---|---|---|---|---|---|---|
| | | (1,1,0) | (1,1,0) | (1,1,$^1$/s) | (1,1,0) | (0.5,1,1) | (0.5) | (-0.25) | (1) | (1.5) |
| [20,16,8] | 92.88±1.05 | **94.53±0.95** | 94.24±0.55 | **94.93±0.60** | 93.55±1.21 | **95.29±0.61** | 92.22±0.83 | **94.59±0.71** | **93.49±1.25** | 93.07±1.08 |
| [20,16,14,12,10,8] | 93.47±0.45 | **94.32±0.94** | **95.11±0.82** | 95.01±0.84 | 94.60±0.70 | **95.31±0.75** | 93.69±0.66 | **94.48±0.58** | 93.93±0.98 | **94.64±0.91** |

Table 4: Accuracy comparison of SPDNet with and without SPD MLRs on the HDM05 dataset.

| Network Architectures | SPDNet | $(\theta, \alpha, \beta)$-AIM | | $(\theta, \alpha, \beta)$-EM | | $(\theta, \alpha, \beta)$-LEM | | $2\theta$-BWM | $\theta$-LCM |
|---|---|---|---|---|---|---|---|---|---|
| | | (1,1,0) | (0.75,1.0,$^1$/$_{30²}$) | (1,1,0) | (0.5,1.0,$^1$/$_{30}$) | (1,1,0) | (0.5,1.0,$^1$/$_{30}$) | (0.5) | (1) |
| [93,30] | 57.42±1.31 | 58.07±0.64 | **59.1±0.59** | 66.32±0.63 | **71.65±0.88** | 56.97±0.61 | **59.30±0.63** | 70.24±0.92 | 48.55±2.35 |
| [93,70,30] | 60.69±0.66 | 60.72±0.62 | **62.18±0.70** | 66.40±0.87 | **70.56±0.39** | 60.69±1.02 | **62.84±0.50** | 70.46±0.71 | 47.61±1.82 |
| [93,70,50,30] | 60.76±0.80 | 61.14±0.94 | **62.36±0.98** | 66.70±1.26 | **70.22±0.81** | 60.28±0.91 | **63.06±0.76** | 70.20±0.91 | 49.10±1.94 |

[3]https://www.dropbox.com/s/dfnlx2bnyh3kjwy/data.zip?dl=0

[4]https://resources.mpi-inf.mpg.de/HDM05/

[5]https://zenodo.org/record/5055046

## 5.1 Experimental Results

For each family of SPD MLRs, we report two representatives: the standard SPD MLR induced from the standard metric ($\theta = 1, \alpha = 1, \beta = 0$), and the one induced from the deformed metric with selected hyper-parameters. Besides, if the standard SPD MLR is already saturated, we report the results of the standard one. In Tabs. 3 to 6, we denote $(\theta, \alpha, \beta)$-AIM as the baseline model endowed with the SPD MLR induced by $(\theta, \alpha, \beta)$-AIM and (1,1,0) as the value of $(\theta, \alpha, \beta)$. So do SPD MLRs under other metrics. We leave the model efficiency in App. F.3.

**Radar:** In line with Brooks et al. (2019), we evaluated our classifiers on the Radar dataset using two network architectures: [20, 16, 8] for the 2-layer configuration and [20, 16, 14, 12, 10, 8] for the 5-layer configuration. The 10-fold results (mean±std) are presented in Tab. 3. Note that as the SPD MLR induced by standard AIM is saturated on this dataset, we report the standard SPD MLR for the family of $(\theta, \alpha, \beta)$-AIM. Generally speaking, our SPD MLRs achieve superior performance against the vanilla LogEig MLR. Moreover, for most families of metrics, the associated SPD MLRs with properly selected hyper-parameters outperform the standard SPD MLR induced by the standard metric, demonstrating the effectiveness of our parameterization. Besides, among all SPD MLRs, the ones induced by $(\theta, \alpha, \beta)$-LEM achieve the best performance on this dataset.

**HDM05:** Following Huang & Van Gool (2017), three architectures are evaluated on this dataset: [93, 30], [93. 70, 30], and [93, 70, 50, 30]. Note that the standard SPD MLRs under BWM and LCM are already saturated on this dataset. Similar observations can be made on this dataset as the Radar dataset. Our SPD MLRs can bring consistent performance gain for SPDNet, and properly selected hyper-parameters can bring further improvement. Particularly, among all the SPD MLRs, the ones based on the $2\theta$-BWM and $\theta$-EM achieved the best performance. These two families of classifiers exhibited **a remarkable increase of approximately 10% accuracy points** compared to the vanilla LogEig MLR, highlighting the effectiveness of our approach. An intriguing aspect is that, despite incompleteness, the $2\theta$-BWM and $\theta$-EM-based classifiers still showed significant performance improvements over the baseline. This result again confirms our theoretical framework's superiority and applicability to a broader range of practical scenarios. However, we observed that the SPD MLRs based on $\theta$-LCM exhibit considerably slower convergence on this dataset. The models fail to converge even after 500 training epochs. We therefore report the results at 500 epochs. This behavior could be attributed to the specific characteristics of the HDM05 dataset, which might interact differently with the $\theta$-LCM metric compared to other metrics.

Table 5: Results of inter-session experiments on the Hinss2021 dataset.

| Methods | SPDDSMBN | $(\theta,\alpha,\beta)$-AIM | | $(\theta,\alpha,\beta)$-EM | $(\theta,\alpha,\beta)$-LEM | | $2\theta$-BWM | $\theta$-LCM | |
|---|---|---|---|---|---|---|---|---|---|
| | | (1,1,0) | (0.5,1,0.05) | (1,1,0) | (1,1,0) | (0.5,1,0.05) | (0.5) | (1) | (1.5) |
| Balanced Acc. | 53.83±9.77 | 53.36±9.92 | **55.27±8.68** | 54.48±9.21 | 53.51±10.02 | **55.26±8.93** | **55.54±7.45** | 55.71±8.57 | **56.43±8.79** |

Table 6: Results of inter-subject experiments on the Hinss2021 dataset.

| Methods | SPDDSMBN | $(\theta,\alpha,\beta)$-AIM | | $(\theta,\alpha,\beta)$-EM | | $(\theta,\alpha,\beta)$-LEM | | $2\theta$-BWM | | $\theta$-LCM | |
|---|---|---|---|---|---|---|---|---|---|---|---|
| | | (1,1,0) | (1.5,1,0) | (1,1,0) | (1.5,1,$^1/_{20}$) | (1,1,0) | (1.25,1,0) | (0.5) | (0.75) | (1) | (0.5) |
| Balanced Acc. | 49.68±7.88 | 50.65±8.13 | **51.15±7.83** | 50.02±5.81 | **51.38±5.77** | 51.41±7.98 | **52.52±6.83** | 50.26±7.23 | **51.67±8.73** | 52.93±7.76 | **54.14±8.36** |

**Hinss2021:** Following Kobler et al. (2022), we adopt the architecture of [40,20]. The results (mean±std) of leaving 5% out cross-validation are reported in Tabs. 5 and 6. Once again, our intrinsic classifiers demonstrate improved performance compared to the baseline, both in the inter-session and inter-subject scenarios. More interestingly, different from the performance on the HDM05 dataset, SPD MLRs based on $\theta$-LCM achieve the best performance (**increase 2.6% for inter-session and 4.46% for inter-subject**), indicating that this metric can faithfully capture the geometry of data in the Hinss2021 dataset. This finding highlights the adaptability and versatility of our framework, as it can effectively leverage different Riemannian metrics based on the intrinsic geometry of the data, leading to improved performance across a wide range of datasets.

## 6 Conclusions

In this paper, we presented a novel and versatile framework for designing intrinsic Riemannian classifiers for matrix manifolds, with a specific focus on SPD manifolds. We systematically explored five families of Riemannian metrics on SPD manifolds and utilized them to construct five families of deformed SPD MLRs. Extensive experiments demonstrated the superiority of our intrinsic classifiers. We expect that our work could present a promising direction for designing intrinsic classifiers in geometric deep learning.

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

APPENDIX CONTENTS

## A  LIMITATIONS AND FUTURE AVENUES

**Limitation:** Recalling our RMLR in Eq. (14), our RMLR is over-parameterized. In RMLR, each class would require an SPD parameter $P_k$ and Euclidean parameter $A_k$. Consequently, as the number of classes grows, the classification layer would become burdened with excessive parameters. We will address this problem in future work.

**Future work:** We highlight the advantage of our approach compared to existing methods is that it only requires the existence of the Riemannian logarithm, which is commonly satisfied by various manifolds encountered in machine learning. Therefore, as a future avenue, our framework offers various possibilities for designing Riemannian classifiers for geometric deep learning on other manifolds.

## B  PRELIMINARIES

### B.1  NOTATIONS

For better understanding, we briefly summarize all the notations used in this paper in Tab. 7.

Table 7: Summary of notations.

| Notation | Explanation |
|---|---|
| $\{\mathcal{M}, g\}$ or abbreviated as $\mathcal{M}$ | A Riemannian manifold |
| $T_P\mathcal{M}$ | The tangent space at $P \in \mathcal{M}$ |
| $g_P(\cdot, \cdot)$ or $\langle \cdot, \cdot \rangle_P$ | The Riemannian metric at $P \in \mathcal{M}$ |
| $\| \cdot \|_P$ | The norm induced by $\langle \cdot, \cdot \rangle_P$ on $T_P\mathcal{M}$ |
| $\mathrm{Log}_P$ | The Riemannian logarithm at $P$ |
| $\Gamma_{P \rightarrow Q}$ | The Riemannian parallel transportation along the geodesic connecting $P$ and $Q$ |
| $H_{a,p}$ | The Euclidean hyperplane |
| $\tilde{H}_{\tilde{A},P}$ | The Riemannian hyperplane |
| $\odot$ | A Lie group operation |
| $\{\mathcal{M}, \odot\}$ | A Lie group |
| $P_{\odot}^{-1}$ | The group inverse of $P$ under $\odot$ |
| $L_P$ | The Lie group left translation by $P \in \mathcal{M}$ |
| $f_{*,P}$ | The differential map of the smooth map $f$ at $P \in \mathcal{M}$ |
| $f^*g$ | The pullback metric by $f$ from $g$ |
| $\mathcal{S}_{++}^n$ | The SPD manifold |
| $\mathcal{S}^n$ | The Euclidean space of symmetric matrices |
| $\mathcal{L}_+^n$ | The Cholesky manifold, *i.e.,* the set of lower triangular matrices with positive diagonal elements |
| $\langle \cdot, \cdot \rangle$ or $\cdot : \cdot$ | The standard Frobenius inner product |
| **ST** | $\mathbf{ST} = \{(\alpha, \beta) \in \mathbb{R}^2 \mid \min(\alpha, \alpha + n\beta) > 0\}$ |
| $\langle \cdot, \cdot \rangle^{(\alpha, \beta)}$ | The $O(n)$-invariant Euclidean inner product |
| $g^{(\alpha, \beta)\text{-LEM}}$ | The Riemannian metric of $(\alpha, \beta)$-LEM |
| $g^{(\alpha, \beta)\text{-AIM}}$ | The Riemannian metric of $(\alpha, \beta)$-AIM |
| $g^{(\alpha, \beta)\text{-EM}}$ | The Riemannian metric of $(\alpha, \beta)$-EM |
| $g^{\text{BWM}}$ | The Riemannian metric of BWM |
| $g^{\text{LCM}}$ | The Riemannian metric of LCM |
| $\log$ | Matrix logarithm |
| Chol | Cholesky decomposition |
| $\mathrm{Dlog}(\cdot)$ | The diagonal element-wise logarithm |
| $\lfloor \cdot \rfloor$ | The strictly lower triangular part of a square matrix |
| $\mathbb{D}(\cdot)$ | A diagonal matrix with diagonal elements from a square matrix |
| $\mathcal{L}_P[\cdot]$ | The Lyapunov operator |
| $(\cdot)^\theta$ | Matrix power |
| $\phi_\theta$ | Power deformation map |

### B.2  BRIEF REVIEW OF RIEMANNIAN GEOMETRY

Intuitively, manifolds are locally Euclidean spaces. Differentials are the generalization of derivatives in classic calculus. For more details on smooth manifolds, please refer to Tu (2011); Lee (2013). Riemannian manifolds are the manifolds endowed with Riemannian metrics, which can be intuitively viewed as point-wise inner products.

**Definition B.1** (Riemannian Manifolds). A Riemannian metric on $\mathcal{M}$ is a smooth symmetric covariant 2-tensor field on $\mathcal{M}$, which is positive definite at every point. A Riemannian manifold is a pair $\{\mathcal{M}, g\}$, where $\mathcal{M}$ is a smooth manifold and $g$ is a Riemannian metric.

W.l.o.g., we abbreviate $\{\mathcal{M}, g\}$ as $\mathcal{M}$. The Riemannian metric $g$ induces various Riemannian operators, including the geodesic, exponential, and logarithmic maps, and parallel transportation. These

operators correspond to straight lines, vector addition, vector subtraction, and parallel displacement in Euclidean spaces, respectively (Pennec et al., 2006, Tabel 1). A plethora of discussions on Riemannian geometry can be found in Do Carmo & Flaherty Francis (1992).

When a manifold $\mathcal{M}$ is endowed with a smooth operation, it is referred to as a Lie group.

**Definition B.2** (Lie Groups). A manifold is a Lie group, if it forms a group with a group operation $\odot$ such that $m(x, y) \mapsto x \odot y$ and $i(x) \mapsto x_{\odot}^{-1}$ are both smooth, where $x_{\odot}^{-1}$ is the group inverse of $x$.

In the main paper, we rely on pullback isometry to study the deformed geometries on SPD manifolds. This idea is a natural generalization of bijection from set theory.

**Definition B.3** (Pullback Metrics). Suppose $\mathcal{M}, \mathcal{N}$ are smooth manifolds, $g$ is a Riemannian metric on $\mathcal{N}$, and $f : \mathcal{M} \to \mathcal{N}$ is smooth. Then the pullback of $g$ by $f$ is defined point-wisely,

$$(f^*g)_p(V_1, V_2) = g_{f(p)}(f_{*,p}(V_1), f_{*,p}(V_2)), \tag{22}$$

where $p \in \mathcal{M}$, $f_{*,p}(\cdot)$ is the differential map of $f$ at $p$, and $V_i \in T_p\mathcal{M}$. If $f^*g$ is positive definite, it is a Riemannian metric on $\mathcal{M}$, which is called the pullback metric defined by $f$.

### B.3 BASIC GEOMETRIES OF SPD MANIFOLDS

In this subsection, we will present necessary Riemannian operators and properties for the five basic geometries on SPD manifolds, namely, $(\alpha, \beta)$-LEM, $(\alpha, \beta)$-EM, $(\alpha, \beta)$-AIM, BWM, and LCM.

For any SPD points $P, Q \in \mathcal{S}_{++}^n$ and tangent vectors $V, W \in T_P\mathcal{S}_{++}^n$, we follow the notations in Tab. 7 and further denote $\tilde{V} = \mathrm{Chol}_{*,P}(V)$, $\tilde{W} = \mathrm{Chol}_{*,P}(W)$, $L = \mathrm{Chol}\, P$, and $K = \mathrm{Chol}\, Q$. For parallel transportation under the BWM, we only present the case where $P, Q$ are commuting matrices, *i.e.*, $P = U\Sigma U^\top$ and $Q = U\Delta U^\top$. We summarize the associated Riemannian operators and properties in Tab. 8.

Table 8: Riemannian operators and properties of five basic metrics on SPD manifolds.

| Name | $g_P(V, W)$ | $\mathrm{Log}_P Q$ | $\Gamma_{P \to Q}(V)$ | Properties |
|---|---|---|---|---|
| $(\alpha, \beta)$-LEM (Thanwerdas & Pennec, 2023) | $\langle \log_{*,P}(V), \log_{*,P}(W) \rangle^{(\alpha,\beta)}$ | $(\log_{*,P})^{-1}[\log(Q) - \log(P)]$ | $(\log_{*,Q})^{-1} \circ \log_{*,P}(V)$ | O($n$)-Invariance, Geodesically Completeness |
| $(\alpha, \beta)$-AIM (Thanwerdas & Pennec, 2023; 2022b) | $\langle P^{-1}V, WP^{-1} \rangle^{(\alpha,\beta)}$ | $P^{1/2} \log\left(P^{-1/2}QP^{-1/2}\right) P^{1/2}$ | $(QP^{-1})^{1/2}V(P^{-1}Q)^{1/2}$ | Lie Group Left-Invariance, O($n$)-Invariance, Geodesically Completeness |
| $(\alpha, \beta)$-EM (Thanwerdas & Pennec, 2023) | $\langle V, W \rangle^{(\alpha,\beta)}$ | $Q - P$ | $V$ | O($n$)-Invariance |
| LCM (Lin, 2019) | $\sum_{i>j} \tilde{V}_{ij}\tilde{W}_{ij} + \sum_{j=1}^n \tilde{V}_{jj}\tilde{W}_{jj}L_{jj}^{-2}$ | $(\mathrm{Chol}^{-1})_{*,L}\left[\lfloor K \rfloor - \lfloor L \rfloor + \mathbb{D}(L)\,\mathrm{Dlog}(\mathbb{D}(L)^{-1}\mathbb{D}(K))\right]$ | $(\mathrm{Chol}^{-1})_{*,K}\left[\lfloor\tilde{V}\rfloor + \mathbb{D}(K)\mathbb{D}(L)^{-1}\mathbb{D}(\tilde{V})\right]$ | Lie Group Bi-Invariance, Geodesically Completeness |
| BWM (Bhatia et al., 2019) | $\frac{1}{2}\langle \mathcal{L}_P[V], W \rangle$ | $(PQ)^{1/2} + (QP)^{1/2} - 2P$ | $U\left[\sqrt{\frac{\delta_i + \delta_j}{\sigma_i + \sigma_j}}\left[U^\top V U\right]_{ij}\right]U^\top$ | O($n$)-Invariance |

### B.4 BASIC LAYERS IN SPDNET AND SPDDSMBN

SPDNet (Huang & Van Gool, 2017) is the most classic SPD neural network. SPDNet mimics the conventional densely connected feedforward network, consisting of three basic building blocks

$$\text{BiMap layer: } S^k = W^k S^{k-1} W^{k\top}, \text{ with } W^k \text{ semi-orthogonal,} \tag{23}$$

$$\text{ReEig layer: } S^k = U^{k-1} \max(\Sigma^{k-1}, \epsilon I_n)U^{k-1\top}, \text{ with } S^{k-1} = U^{k-1}\Sigma^{k-1}U^{k-1\top}, \tag{24}$$

$$\text{LogEig layer: } S^k = \log(S^{k-1}). \tag{25}$$

where $\max()$ is element-wise maximization. BiMap and ReEig mimic transformation and non-linear activation, while LogEig maps SPD matrices into the tangent space at the identity matrix for classification.

SPDNetBN (Brooks et al., 2019) further proposed Riemannian batch normalization based on AIM:

$$\text{Centering from geometric mean } \mathfrak{G} : \forall i \leq N, \bar{S}_i = \mathfrak{G}^{-\frac{1}{2}} S_i \mathfrak{G}^{-\frac{1}{2}}, \tag{26}$$

$$\text{Biasing towards SPD parameter } G : \forall i \leq N, \tilde{S}_i = G^{\frac{1}{2}} \bar{S}_i G^{\frac{1}{2}}. \tag{27}$$

SPD domain-specific momentum batch normalization (SPDDSMBN) is an improved version of SPDNetBN. Apart from controlling mean, it also can control variance. The key operation in SPDDSMBN of controlling mean and variance is:

$$\Gamma_{I \to G} \circ \Gamma_{\mathfrak{G} \to I}(S_i)^{\frac{\nu}{\nu + \varepsilon}}, \tag{28}$$

where $\mathfrak{G}$ and $\bar{v}$ are Riemannian mean and variance. Inspired by Yong et al. (2020), during the training stage, SPDDSMBN generates running means and running variances for training and testing with distinct momentum parameters. Besides, it sets $\mathfrak{G}$ and $\bar{v}$ as the running mean and running variance w.r.t. training for training and the ones w.r.t. testing for testing. SPDDSMBN also applies domain-specific techniques (Chang et al., 2019), keeping multiple parallel BN layers and distributing observations according to the associated domains. To crack cross-domain knowledge, $v$ is uniformly learned across all domains, and $G$ is set to be the identity matrix.

## C  RIEMANNIAN MLR AS A NATURAL EXTENSION OF EUCLIDEAN MLR

**Proposition C.1.** *When $\mathcal{M} = \mathbb{R}^n$ is the standard Euclidean space, the Riemannian MLR defined in Thm. 3.4 becomes the Euclidean MLR in Eq.* (3).

*Proof.* On the standard Euclidean space $\mathbb{R}^n$, $\mathrm{Log}_y x = x - y, \forall x, y \in \mathbb{R}^n$. Besides, the differential maps of left translation and parallel transportation are the identity maps. Therefore, given $x, p_k \in \mathbb{R}^n$ and $a_k \in \mathbb{R}^n / \{0\} \cong T_0 \mathbb{R}^n / \{0\}$, we have

$$p(y = k \mid x \in \mathbb{R}^n) \propto \exp(\langle \mathrm{Log}_{p_k} x, a_k \rangle_{p_k}), \tag{29}$$

$$\propto \exp(\langle x - p_k, a_k \rangle), \tag{30}$$

$$\propto \exp(\langle x, a_k \rangle - b_k), \tag{31}$$

where $b_k = \langle x, p_k \rangle$. $\qquad\square$

## D  THEORIES ON THE DEFORMED METRICS

### D.1  PROPERTIES OF THE DEFORMED METRICS (TAB. 1)

In this subsection, we prove the properties presented in Tab. 1.

*Proof.* Firstly, we prove $O(n)$-invariance of $(\theta, \alpha, \beta)$-LEM, $(\theta, \alpha, \beta)$-EM, $(\theta, \alpha, \beta)$-AIM, and $2\theta$-BWM. Since the differential of $\phi_\theta$ is $O(n)$-equivariant, and $(\alpha, \beta)$-LEM, $(\alpha, \beta)$-EM, $(\alpha, \beta)$-AIM, and BWM are $O(n)$-invariant (Thanwerdas & Pennec, 2023), $O(n)$-invariance are thus acquired.

Next, we focus on geodesic completeness. It can be easily proven that Riemannian isometries preserve geodesic completeness. On the other hand, $(\alpha, \beta)$-LEM, $(\alpha, \beta)$-AIM, and LCM are geodesically complete (Thanwerdas & Pennec, 2023; Lin, 2019). As a direct corollary, geodesic completeness can be obtained since $\phi_\theta$ is a Riemannian isometry.

Finally, we deal with Lie group invariance. Similarly, it can be readily proved that Lie group invariance is preserved under isometries. LCM, LEM, and $(\alpha, \beta)$-AIM are Lie group bi-invariant (Lin, 2019), bi-invariant (Arsigny et al., 2005), and left-invariant (Thanwerdas & Pennec, 2022b). As an isometric pullback metric from the standard LEM (Thanwerdas & Pennec, 2023), $(\alpha, \beta)$-LEM is, therefore, Lie group bi-invariant. As pullback metrics, $(\theta, \alpha, \beta)$-LEM, $(\theta, \alpha, \beta)$-AIM, and $\theta$-LCM are therefore bi-invariant, left-invariant, and bi-invariant, respectively. $\qquad\square$

### D.2  LIMITING CASES OF THE DEFORMED METRICS (FIG. 1)

In this subsection, we prove the limiting cases in Fig. 1. In detail, we need to prove the following cases under $\theta \to 0$:

(a) $2\theta$-BWM tends to be $(\frac{1}{4}, 0)$-LEM;

(b) $(\theta, \alpha, \beta)$-EM tends to be $(\alpha, \beta)$-LEM;

(c) $(\theta, \alpha, \beta)$-LEM tends to be $(\alpha, \beta)$-LEM;

(d) $\theta$-LCM tends to be $\tilde{g}$-LEM (defined in Thm. 4.2).

Before starting the proof, we first recall a well-known property of deformed metrics (Thanwerdas & Pennec, 2022a).

**Lemma D.1.** *Let $\phi_\theta^* g$ be the deformed metric on SPD manifolds pulled back from $g$ by the power deformation map $\phi_\theta$. Then when $\theta$ tends to 0, for all $P \in \mathcal{S}_{++}^n$ and all $V \in T_P \mathcal{S}_{++}^n$, we have*

$$(\phi_\theta^* g)_P(V, V) \to g_I(\log_{*,P}(V), \log_{*,P}(V)). \tag{32}$$

Now, we present our proof for the limiting cases of deformed metrics.

*Proof.* By Lem. D.1, we only have to compute $g_I(\cdot, \cdot)$ ($g$ could be $(\alpha, \beta)$-EM, $(\alpha, \beta)$-LEM, LCM, and BWM). Simple computations show that

$$g_I^{(\alpha,\beta)\text{-EM}}(V, V) = \langle V, V \rangle^{(\alpha,\beta)}, \tag{33}$$

$$g_I^{(\alpha,\beta)\text{-LEM}}(V, V) = \langle V, V \rangle^{(\alpha,\beta)}, \tag{34}$$

$$g_I^{\text{LCM}}(V, V) = \tilde{g}(V, V), \tag{35}$$

$$g_I^{\text{BWM}}(V, V) = \frac{1}{4}\langle V, V \rangle^{(\alpha,\beta)}. \tag{36}$$

Together with Eq. (32), one can directly get all the results. □

# E  TECHNICAL DETAILS ON THE PROPOSED SPD MLRs

## E.1  COMPUTING MATRIX SQUARE ROOTS IN SPD MLRs UNDER POWER BWMs

In the case of MLRs induced by $2\theta$-BWM, computing square roots like $(BA)^{\frac{1}{2}}$ and $(AB)^{\frac{1}{2}}$ with $B, A \in \mathcal{S}_{++}^n$ poses a challenge. Eigendecomposition cannot be directly applied, since $BA$ and $AB$ are no longer symmetric, let alone positive definity. Instead, we use the following formulas to compute these square roots (Minh, 2022):

$$(BA)^{\frac{1}{2}} = B^{\frac{1}{2}}(B^{\frac{1}{2}}AB^{\frac{1}{2}})^{\frac{1}{2}}B^{-\frac{1}{2}} \text{ and } (AB)^{\frac{1}{2}} = [(BA)^{\frac{1}{2}}]^\top, \tag{37}$$

where the involved square roots can be computed using eigendecomposition or singular value decomposition (SVD).

## E.2  NUMERICAL STABILITY OF SPD MLRs UNDER POWER BWMs

Let us first explain why we abandon parallel transportation on the SPD MLR derived from $2\theta$-BWM. Then, we propose our numerically stable methods for computing the SPD MLR based on $2\theta$-BWM.

### E.2.1  INSTABILITY OF PARALLEL TRANSPORTATION UNDER POWER BWMs

As discussed in Thm. 3.4, there are two ways to generate $\tilde{A}$ in SPD MLR: parallel transportation and Lie group translation. However, parallel transportation under $2\theta$-BWM could cause numerical problems. W.l.o.g., we focus on the standard BWM as $2\theta$-BWM is isometric to the BWM.

Although the general solution of parallel transportation under BWM is the solution of an ODE, for the case of parallel transportation starting from the identity matrix, we have a closed-form expression (Thanwerdas & Pennec, 2023):

$$\Gamma_{I \to P}(V) = U \left[ \sqrt{\frac{\sigma_i + \sigma_j}{2}} \left[ U^\top V U \right]_{ij} \right] U^\top, \tag{38}$$

where $P = U\Sigma U^\top$ is the eigendecomposition of $P \in \mathcal{S}_{++}^n$. There would be no problem in the forward computation of Eq. (38). However, during backpropagation (BP), Eq. (38) would require the BP of eigendecomposition, involving the calculation of $1/(\sigma_i - \sigma_j)$ (Ionescu et al., 2015, Prop. 2). When $\sigma_i$ is close to $\sigma_j$, the BP of eigendecomposition could be problematic.

### E.2.2  NUMERICALLY STABLE METHODS FOR SPD MLRs BASED ON POWER BWMs

To bypass the instability of parallel transportation under BWM, we use Lie group left translation to generate $\tilde{A}$ in MLRs induced from $2\theta$-BWM. However, there is another problem that could cause instability. The computation of the Riemannian metric of $2\theta$-BWM requires solving the Lyapunov operator, *i.e.,* $\mathcal{L}_P[V]P + P\mathcal{L}_P[V] = V$. Under the case of symmetric matrices, the Lyapunov operator can be obtained by eigendecomposition:

$$\mathcal{L}_P[V] = U \left[ \frac{V'_{ij}}{\sigma_i + \sigma_j} \right]_{i,j} U^\top, \tag{39}$$

where $V \in \mathcal{S}^n$, $UV'U^\top = V$, and $P = U\Sigma U^\top$ is the eigendecomposition of $P \in \mathcal{S}_{++}^n$. Similar with Eq. (38), the BP of Eq. (39) requires $1/(\sigma_i - \sigma_j)$, undermining the numerical stability.

To remedy this problem, we proposed the following formula to stably compute the BP of Eq. (39).

**Proposition E.1.** *For all $P \in \mathcal{S}_{++}^n$ and all $V \in \mathcal{S}^n$, we denote the Lyapunov equation as*

$$XP + PX = V, \tag{40}$$

*where $X = \mathcal{L}_P[V]$. Given the gradient $\frac{\partial L}{\partial X}$ of loss $L$ w.r.t. $X$, then the BP of the Lyapunov operator can be computed by:*

$$\frac{\partial L}{\partial V} = \mathcal{L}_P[\frac{\partial L}{\partial X}], \tag{41}$$

$$\frac{\partial L}{\partial P} = -X\mathcal{L}_P[\frac{\partial L}{\partial X}] - \mathcal{L}_P[\frac{\partial L}{\partial X}]X, \tag{42}$$

*where $\mathcal{L}_P[\cdot]$ can be computed by Eq. (39).*

*Proof.* Differentiating both sides of Eq. (40), we obtain

$$\mathrm{d}\, XP + X\,\mathrm{d}\, P + \mathrm{d}\, PX + P\,\mathrm{d}\, X = \mathrm{d}\, V, \tag{43}$$

$$\implies \mathrm{d}\, XP + P\,\mathrm{d}\, X = \mathrm{d}\, V - X\,\mathrm{d}\, P - \mathrm{d}\, PX, \tag{44}$$

$$\implies \mathrm{d}\, X = \mathcal{L}_P[\mathrm{d}\, V - X\,\mathrm{d}\, P - \mathrm{d}\, PX]. \tag{45}$$

Besides, easy computations show that

$$\mathcal{L}_P[V] : W = V : \mathcal{L}_P[W], \forall W, V \in \mathcal{S}^n, \tag{46}$$

where $\cdot : \cdot$ denotes the standard Frobenius inner product.

Then we have the following:

$$\frac{\partial L}{\partial X} : \mathrm{d}\, X = \frac{\partial L}{\partial X} : \mathcal{L}_P[\mathrm{d}\, V - X\,\mathrm{d}\, P - \mathrm{d}\, PX], \tag{47}$$

$$\implies \frac{\partial L}{\partial X} : \mathrm{d}\, X = \mathcal{L}_P[\frac{\partial L}{\partial X}] : \mathrm{d}\, V + \left( -X\mathcal{L}_P[\frac{\partial L}{\partial X}] - \mathcal{L}_P[\frac{\partial L}{\partial X}]X \right) : \mathrm{d}\, P. \tag{48}$$

$$\square$$

*Remark* E.2. Eq. (39) needs to be computed in the Lyapunov operator's forward and backward process. Therefore, in the forward process, we can save the intermediate matrices $U$ and $K$ with $K_{i,j} = \left[ \frac{1}{\sigma_i + \sigma_j} \right]_{i,j}$, and then use them to compute the backward process efficiently.

### E.3 DETAILS ON FIVE FAMILIES OF DEFORMED SPD MLRS (TAB. 2)

In this subsection, we will apply Lem. 4.3 to derive the expressions of our SPD MLRs presented in Tab. 2. For our cases of SPD MLRs, we set $Q = I$. W.l.o.g., we will omit the subscript $k$ for $P_k$ and $A_k$ In the following proof, we will first derive the expressions of SPD MLRs under $(\theta, \alpha, \beta)$-LEM, $(\theta, \alpha, \beta)$-AIM, $(\theta, \alpha, \beta)$-EM, and $\theta$-LCM, as they are sourced from Eq. (19). Then we will proceed to present the expression of MLR under $2\theta$-BWM, which is sourced from Eq. (21).

*Proof.* For simplicity, we abbreviate $\phi_\theta$ as $\phi$ during the proof. Note that for $2\theta$-BWM, $\phi$ should be understood as $\phi_{2\theta}$. We first show $\phi(I)$ and differential map $\phi_{*,I}$, which will be frequently required in the following proof:

$$\phi(I) = \frac{1}{|\theta|}I, \tag{49}$$

$$\phi_{*,I}(A) = \mathrm{sgn}(\theta)(A), \forall A \in T_I \mathcal{S}_{++}^n. \tag{50}$$

Then we can showcase Eq. (19) on SPD manifolds with $Q = I$. Denoting $\phi : \{\mathcal{S}_{++}^n, \tilde{g}\} \to \{\mathcal{S}_{++}^n, g\}$, then the SPD MLR under $\tilde{g}$ by parallel transportation is

$$p(y = k \mid S \in \mathcal{M}) = \exp\left[ g_{\phi(P)}(\mathrm{Log}_{\phi(P)} \phi(S), \Gamma_{\frac{1}{|\theta|}I \to \phi(P)} \mathrm{sgn}(\theta)(A)) \right], \tag{51}$$

Next, we begin to prove the five SPD MLRs one by one.

$(\theta, \alpha, \beta)$**-LEM:** Obviously, $(\theta, \alpha, \beta)$-LEM is the pullback metric from the Euclidean space $\{\mathcal{S}^n, g^{(\alpha, \beta)}\}$:

$$\{\mathcal{S}_{++}^n, g^{(\theta, \alpha, \beta)\text{-LEM}}\} \xrightarrow{\phi} \{\mathcal{S}_{++}^n, g^{(\alpha, \beta)\text{-LEM}}\} \xrightarrow{\log} \{\mathcal{S}^n, g^{(\alpha, \beta)}\} \tag{52}$$

Denote $\psi = \log \circ \phi$. For the differential of $\psi$, we have

$$\psi_{*, I}(A) = \theta A, \forall A \in T_I \mathcal{S}_{++}^n, \tag{53}$$

Putting Eq. (53) and $g^{(\alpha, \beta)}$ into Eq. (19), we have

$$p(y = k \mid S \in \mathcal{M}) \propto \exp \left[ \langle \psi(S) - \psi(P), \theta A \rangle^{(\alpha, \beta)} \right], \tag{54}$$

$$= \exp \left[ \langle \log(S^\theta) - \log(P^\theta), \theta A \rangle^{(\alpha, \beta)} \right], \tag{55}$$

$$= \exp \left[ \theta^2 \langle \log(S) - \log(P), A \rangle^{(\alpha, \beta)} \right], \tag{56}$$

We showcase this process for MLRs under $(\theta, \alpha, \beta)$-LEM.

$(\theta, \alpha, \beta)$**-AIM:** Putting $g^{(\alpha, \beta)\text{-AIM}}$ into Eq. (51), we have

$$p(y = k \mid S \in \mathcal{M}) \propto \exp \left[ g_{\phi(P)}^{(\alpha, \beta)\text{-AIM}} (\frac{1}{|\theta|} P^{\frac{\theta}{2}} \log(P^{-\frac{\theta}{2}} S^\theta P^{-\frac{\theta}{2}}) P^{\frac{\theta}{2}}, P^{\frac{\theta}{2}} \operatorname{sgn}(\theta) A P^{\frac{\theta}{2}}) \right], \tag{57}$$

$$= \exp \left[ \langle \log(P^{-\frac{\theta}{2}} S^\theta P^{-\frac{\theta}{2}}), \theta A \rangle^{(\alpha, \beta)} \right]. \tag{58}$$

$(\theta, \alpha, \beta)$**-EM:** Putting $g^{(\alpha, \beta)\text{-EM}}$ into Eq. (51), we have

$$p(y = k \mid S \in \mathcal{M}) \propto \exp \left[ g_{\phi(P)}^{(\alpha, \beta)\text{-EM}} (\phi(S) - \phi(P), \operatorname{sgn}(\theta) A) \right], \tag{59}$$

$$= \exp \left[ \frac{1}{\theta} \langle S^\theta - P^\theta, A \rangle^{(\alpha, \beta)} \right]. \tag{60}$$

$\theta$**-LCM**: Simple computation shows that $\theta$-LCM is the pullback metric of standard Euclidean metric in $\mathcal{S}^n$:

$$\{\mathcal{S}_{++}^n, g^{\theta\text{-LCM}}\} \xrightarrow{\phi} \{\mathcal{S}_{++}^n, g^{\text{LCM}}\} \xrightarrow{\text{Chol}} \{\mathcal{L}_+^n, g^{\text{CM}}\} \xrightarrow{\text{Dlog}} \{\mathcal{S}^n, g^{\text{E}}\}, \tag{61}$$

where $g^{\text{E}}$ is the standard Frobenius inner product, and $g^{\text{CM}}$ is the Cholesky metric on the Cholesky space $\mathcal{L}_+^n$ (Lin, 2019). We denote $\zeta = \text{Dlog} \circ \text{Chol} \circ \phi$, then we have

$$\zeta_{*, I}(A) = \operatorname{sgn}(\theta)(\sqrt{|\theta|} \lfloor A \rfloor + \frac{|\theta|}{2} \mathbb{D}(A)), \tag{62}$$

$$\zeta(I) = -\log(\sqrt{|\theta|}) I. \tag{63}$$

Similar with the case of $(\theta, \alpha, \beta)$-LEM, we have

$$p(y = k \mid S \in \mathcal{M}) \tag{64}$$

$$\propto \exp \left[ \langle \zeta(S) - \zeta(P), \zeta_{*, I} A \rangle \right], \tag{65}$$

$$= \exp \left[ \operatorname{sgn}(\theta) \langle \lfloor \tilde{K} \rfloor - \lfloor \tilde{L} \rfloor + \sqrt{|\theta|} \left[ \text{Dlog}(\mathbb{D}(\tilde{K})) - \text{Dlog}(\mathbb{D}(\tilde{L})) \right], \lfloor A \rfloor + \frac{\sqrt{|\theta|}}{2} \mathbb{D}(A) \rangle \right], \tag{66}$$

where $\tilde{K} = \text{Chol}(S^\theta)$, $\tilde{L} = \text{Chol}(P^\theta)$, $\mathbb{D}(\tilde{K})$ is a diagonal matrix with diagonal elements from $\tilde{K}$, and $\lfloor \tilde{K} \rfloor$ is a strictly lower triangular matrix from $\tilde{K}$.

$2\theta$**-BWM:** We first simplify Eq. (21) under the cases of SPD manifolds and then proceed to focus on the case of $g = g^{\text{BWM}}$. Denote $\phi : \{\mathcal{S}_{++}^n, \tilde{g}, \tilde{\odot}\} \rightarrow \{\mathcal{S}_{++}^n, g, \odot\}$, where the Lie group operation $\odot$ (Thanwerdas & Pennec, 2022b) is defined as

$$S_1 \odot S_2 = L_1 S_2 L_1^T, \forall S_1, S_2 \in \mathcal{S}_{++}^n, \text{ with } L_1 = \text{Chol}(S_1). \tag{67}$$

Note that $I$ is the identity element of $\{\mathcal{S}_{++}^n, \odot\}$, and for any $S \in \mathcal{S}_{++}^n$, the differential map of the left translation $L_S$ under $\odot$ is

$$L_{S*,Q}(V) = LVL^\top, \forall Q \in \mathcal{S}_{++}^n, \forall V \in T_Q\mathcal{S}_{++}^n, \text{ where } L = \text{Chol}(S). \tag{68}$$

For the induced Lie group $\{\mathcal{S}_{++}^n, \tilde{\odot}\}$, the left translation $\tilde{L}_{P\tilde{\odot}I_{\tilde{\odot}}^{-1}}$ under $\tilde{\odot}$ is

$$\tilde{L}_{P\tilde{\odot}I_{\tilde{\odot}}^{-1}} = \phi^{-1} \circ L_{\phi(P)\odot\phi(I)_\odot^{-1}} \circ \phi, \tag{69}$$

$$= \phi^{-1} \circ L_{P^{2\theta}} \circ \phi. \quad (\phi(P) \odot \phi(I)_\odot^{-1} = P^{2\theta}) \tag{70}$$

The associated differential at $I$ is

$$\tilde{L}_{P\tilde{\odot}I_{\tilde{\odot}}^{-1}*,I}(A) = \phi_{*,\phi(P)}^{-1} \circ L_{P^{2\theta}*,\phi(I)} \circ \phi_{*,I}(A), \tag{71}$$

$$= \text{sgn}(\theta)\phi_{*,\phi(P)}^{-1}(\bar{L}A\bar{L}^\top), \tag{72}$$

where $\bar{L} = \text{Chol}(P^{2\theta})$. Then the SPD MLRs under $\tilde{g}$ and $\tilde{\odot}$ by left translation is

$$p(y = k \mid S \in \mathcal{M}) = \exp\left[\text{sgn}(\theta)g_{\phi(P)}\left(\text{Log}_{\phi(P)}\phi(S), \bar{L}A\bar{L}^\top\right)\right], \tag{73}$$

Setting $g = g^{\text{BWM}}$, we obtain the SPD MLR under $2\theta$-BWM:

$$p(y = k \mid S \in \mathcal{M}) = \exp\left[\text{sgn}(\theta)g_{\phi(P)}^{\text{BWM}}\left(\text{Log}_{\phi(P)}^{\text{BWM}}\phi(S), \bar{L}A\bar{L}^\top\right)\right], \tag{74}$$

$$= \exp\left[\frac{\text{sgn}(\theta)}{2}\langle(P^{2\theta}S^{2\theta})^{\frac{1}{2}} + (S^{2\theta}P^{2\theta})^{\frac{1}{2}} - 2P^{2\theta}, \mathcal{L}_{P^{2\theta}}(\bar{L}A\bar{L}^\top)\rangle\right]. \tag{75}$$

$\square$

## F    EXPERIMENTAL DETAILS

### F.1    HYPER-PARAMETERS

For the deformation factor $\theta$, we roughly select its value around its deformation boundary, *i.e.*,[0.25,1.5] for $(\theta, \alpha, \beta)$-AIM and $(\theta, \alpha, \beta)$-LEM (Recalling Tab. 2, for $(\theta, \alpha, \beta)$-LEM, w.l.o.g., $\theta$ is positive), [0.5,1.5] for $\theta$-LCM, [0.25,1.5] and $(\theta, \alpha, \beta)$-EM, [-0.75,0.75] for $2\theta$-BWM. We equally select several candidate values in each deformation interval. The details values are listed in Tab. 9.

Table 9: Candidate values for hyper-parameters in SPD MLRs

| Metric | $(\theta, \alpha, \beta)$-AIM | $(\theta, \alpha, \beta)$-LEM | $(\theta, \alpha, \beta)$-EM | $\theta$-LCM | $2\theta$-BWM |
|---|---|---|---|---|---|
| Candidate Values | { 0.25,0.5,0.75,1,1.25,1.5 } | {0.25,0.5,0.75,1,1.25,1.5} | {0.5,1,1.5 } | {0.5,1,1.5 } | {±0.75,±0.5,±0.25 } |

### F.2    PREPROCESSING OF THE HINSS2021 DATASET

We follow the Python implementation[6] (Kobler et al., 2022) to carry out preprocessing. In detail, the python package MOABB (Jayaram & Barachant, 2018) and MNE (Gramfort, 2013) are used to preprocess the datasets. The applied steps include resampling the EEG signals to 250/256 Hz, applying temporal filters to extract oscillatory EEG activity in the 4 to 36 Hz range, extracting short segments ( $\leq$ 3s) associated with a class label, and finally obtaining $40 \times 40$ SPD covariance matrices.

### F.3    MODEL EFFICIENCY

We adopt the deepest architectures, namely [20, 16, 14, 12, 10, 8] for the Radar dataset, [93, 70, 50, 30] for the HDM05 dataset, and [40, 20] for the Hinss2021 dataset. For simplicity, we focus on the SPD MLRs induced by standard metrics, *i.e.*, AIM, EM, LEM, BWM, and LCM. The average training time (in seconds) per epoch is reported in Tab. 10. Generally, when the number of classes is small (*e.g.*, 3 in the Radar and Hinss2021 datasets), our SPD MLRs only bring minor additional

---

[6]https://github.com/rkobler/TSMNet

Table 10: Training efficiency.

| Methods | Radar | HDM05 | Hinss2021 | |
| --- | --- | --- | --- | --- |
| | | | Inter-session | Inter-subject |
| Baseline | 1.36 | 1.95 | 0.18 | 8.31 |
| AIM-MLR | 1.75 | 31.64 | 0.38 | 13.3 |
| EM-MLR | 1.34 | 3.91 | 0.19 | 8.23 |
| LEM-MLR | 1.5 | 4.7 | 0.24 | 10.13 |
| BWM-MLR | 1.75 | 33.14 | 0.38 | 13.84 |
| LCM-MLR | 1.35 | 3.29 | 0.18 | 8.35 |

training time compared to the baseline LogEig MLR. However, when dealing with a larger number of classes (*e.g.,*117 classes in the HDM05 dataset), there could be some inefficiency caused by our SPD MLRs. This is because each class requires an SPD parameter, and each parameter might require matrix decomposition in the forward or backward processes during training. Nonetheless, the SPD MLRs induced by EM or LCM generally achieve comparable efficiency with the vanilla LogEig MLR. This is due to the fast computation of their Riemannian operators, making them efficient choices for tasks with a larger number of classes. This result highlights the flexibility of our framework and its applicability to various scenarios.

### F.4 VISUALIZATION

This subsection visualizes the 10-fold average results of SPDNet with different classifiers on the Radar and HDM05 datasets. We focus on the deepest architectures, *i.e.,*. [20,16,14,12,10,8] for the Radar dataset, and [93,70,50,30] for the HDM05 dataset. Note that we only report the SPD MLR with the best hyper-parameters $(\theta, \alpha, \beta)$. The figures are presented in Fig. 3. All the results are sourced from Tabs. 3 and 4.

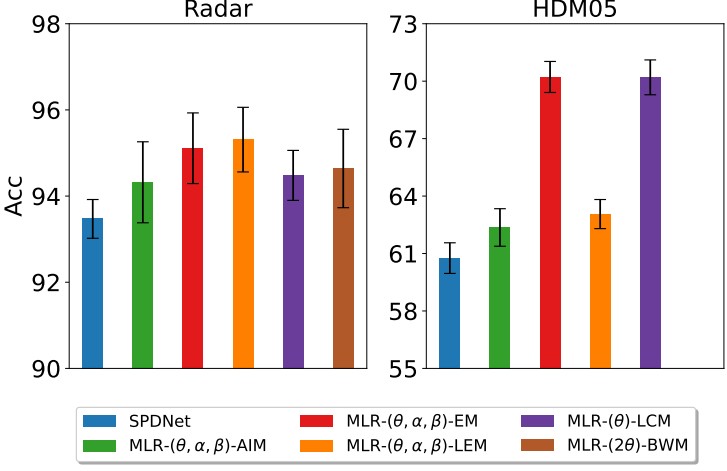

Figure 3: Visualization of 10-fold average accuracy of SPDNet with different SPD MLRs on the Radar and HDM05 datasets. The error bar denotes the standard deviation.

## G DIFFERENCE OF OUR SPD MLRS WITH THE EXISTING SPD MLRS

The main paper showcases our RMLR on SPD manifolds under five families of deformed metrics. Our SPD MLRs incorporate the MLRs presented in Chen et al. (2023a); Nguyen & Yang (2023). Besides, none of them develop SPD MLRs under BWM or Euclidean Metric (EM). In contrast, we systematically discuss five deformed families of SPD MLRs. We briefly summarize the difference in Tab. 11.

Table 11: Difference between our SPD MLRs and the previous SPD MLRs.

| SPD MLR | Metrics Involved | Theoretical Foundations | Geometric Requirement |
|---|---|---|---|
| Gyro SPD MLRs in (Nguyen & Yang, 2023) | The standard AIM, LEM, and LCM | Gyro-structures induced by the standard AIM, LEM, and LCM | Gyro-structures, geodesic completeness |
| Flat SPD MLRs in (Chen et al., 2023a) | $(\alpha, \beta)$-LEM | Pullback metric from the Euclidean space | Pullback metric from the Euclidean space |
| Ours | $(\theta, \alpha, \beta)$-AIM, $(\theta, \alpha, \beta)$-LEM, $(\theta, \alpha, \beta)$-EM, $\theta$-LCM, $2\theta$-BWM | Riemannian geometry | Riemannian logarithm |

# H  PROOFS OF THE LEMMAS AND THEORIES IN THE MAIN PAPER

*Proof of Thm. 3.3.* Let us first solve $Q^*$ in Eq. (11), which is the solution to the following constrained optimization problem:

$$\max_{Q} \left( \frac{\langle \mathrm{Log}_P Q, \mathrm{Log}_P S \rangle_P}{\| \mathrm{Log}_P Q \|_P, \| \mathrm{Log}_P S \|_P} \right) \quad \text{s.t.} \langle \mathrm{Log}_P S, \tilde{A} \rangle_P = 0 \tag{76}$$

Note that Eq. (76) is well-defined due to the existence of logarithm. Although, Eq. (76) is normally non-convex, Eq. (76) and Eq. (11) can be reduced to a Euclidean problem:

$$\max_{\tilde{Q}} \frac{\langle \tilde{Q}, \tilde{S} \rangle_P}{\| \tilde{Q} \|_P \| \tilde{S} \|_P} \quad \text{s.t.} \langle \tilde{Q}, \tilde{A} \rangle_P = 0, \tag{77}$$

$$d(S, \tilde{H}_{\tilde{A}, P}) = \sin(\angle SPQ^*) \| \tilde{S} \|_P, \tag{78}$$

where $\tilde{Q} = \mathrm{Log}_P Q$ and $\tilde{S} = \mathrm{Log}_P S$.

Let us first discuss Eq. (77). Denote the solution of Eq. (77) as $\tilde{Q}^*$. Note that $\tilde{Q}^*$ is not necessarily unique. Note that $\mathrm{Exp}_P$ is only well-defined locally. More precisely, $\mathrm{Exp}_P$ is well-defined in an open ball $\mathrm{B}_\epsilon(0)$ centered at $0 \in T_P\mathcal{M}$. Therefore, $\tilde{Q}^*$ might not be in $\mathrm{B}_\epsilon(0)$. In this case, we can scale $\tilde{Q}^*$ into $\mathrm{B}_\epsilon(0)$, and the scaled $\tilde{Q}^*$ is still the maximizer of Eq. (77). Therefore, w.l.o.g., we assume $\tilde{Q}^* \in \mathrm{B}_\epsilon(0)$.

Putting $\tilde{Q}^*$ into Eq. (78), Eq. (78) is reduced to the distance to the hyperplane $\langle \tilde{Q}, \tilde{A} \rangle_P = 0$ in the Euclidean space $\{T_P\mathcal{M}, \langle \cdot, \cdot \rangle_P\}$, which has a closed-form solution:

$$d(S, \tilde{H}_{\tilde{A}, P}) = \frac{|\langle \tilde{S}, \tilde{A} \rangle_P|}{\| \tilde{A} \|_P}, \tag{79}$$

$$= \frac{|\langle \mathrm{Log}_P S, \tilde{A} \rangle_P|}{\| \tilde{A} \|_P}. \tag{80}$$

$\square$

*Proof for Thm. 3.4.* Putting the margin distance (Eq. (13)) into Eq. (6), RMLR can be obtained. $\square$

*Proof for Thm. 4.2.* Please refer to App. D.2. $\square$

*Proof for Lem. 4.3.* Before starting, we should point out that since $\phi$ is a diffeomorphism, $\tilde{\odot}$ and $\tilde{g}$ are indeed well defined, and $\{\mathcal{M}, \tilde{g}\}$ forms a Riemannian manifold and $\{\mathcal{M}, \tilde{\odot}\}$ forms a Lie group (Chen et al., 2023b, Lemma 3.2). We denote $\phi_*^{-1}$ as the differential of $\phi^{-1}$. We first focus on the Riemannian MLR by parallel transportation:

$$p(y = k \mid S \in \mathcal{M}) \tag{81}$$

$$\propto \exp(\tilde{g}_{P_k}(\tilde{\mathrm{Log}}_{P_k} S, \tilde{\Gamma}_{Q \to P_k} A_k)), \tag{82}$$

$$= \exp\left[ g_{\phi(P_k)} \left( \phi_{*,P_k} \circ \phi_{*,\phi(P_k)}^{-1} \mathrm{Log}_{\phi(P_k)} \phi(S), \phi_{*,P_k} \circ \phi_{*,\phi(P_k)}^{-1} \Gamma_{\phi(Q) \to \phi(P_k)} \phi_{*,Q}(A_k) \right) \right], \tag{83}$$

$$= \exp\left[ g_{\phi(P_k)}(\mathrm{Log}_{\phi(P_k)} \phi(S), \Gamma_{\phi(Q) \to \phi(P_k)} \phi_{*,Q}(A_k)) \right]. \tag{84}$$

In the case of the Riemannian MLR by left translation, we first note that:

$$\tilde{L}_{\tilde{R}_k} = \phi^{-1} \circ L_{\phi(P_k) \odot \phi(Q)_\odot^{-1}} \circ \phi. \tag{85}$$

Therefore, the associated differential is:

$$\tilde{L}_{\tilde{R}_k *} = \phi_*^{-1} \circ L_{\phi(P_k) \odot \phi(Q)_\odot^{-1} *} \circ \phi_*. \tag{86}$$

Putting Eq. (86) in Eq. (20), we can obtain the results. □

