# OpenReview forum: "Intrinsic Riemannian Classifiers on the Deformed SPD Manifolds: A Unified Framework"
_ICLR.cc/2024/Conference — Submitted to ICLR 2024_

### Official Review · Reviewer_ur4M · 2023-10-18

**Soundness:** 2 fair
**Presentation:** 2 fair
**Contribution:** 2 fair
**Rating:** 6
**Confidence:** 3

**Summary:**

Geometric deep learning has gained attention for extending deep learning to non-Euclidean spaces. To improve the classification of non-Euclidean features, researchers have explored intrinsic classifiers based on Riemannian geometry. However, existing approaches are limited due to their reliance on specific geometric properties. This paper introduces a general framework for designing multinomial logistic regression on Riemannian manifolds. This framework requires minimal geometric properties. The focus is on symmetric positive definite (SPD) manifolds, and the study includes five families of parameterized Riemannian metrics to develop diverse SPD classifiers. The versatility and effectiveness of this framework are demonstrated in applications such as radar recognition, human action recognition, and EEG classification.

**Strengths:**

The paper addresses the problem of supervised classification on Riemannian manifolds with a focus on the SPD manifold. The latter is used extensively to classify biosignals such as MEG or EEG.
Several applications are considered: classification of radar, human action and EEG data.
The deformation $\theta$ shows promising results in application and can be used in placed in many classical classification algorithms.

**Weaknesses:**

The paper is quite hard to follow.

First of all, the authors claim their approach is general in terms of classifiers and Riemannian manifolds. However, they only derive results for multinomial logistic regression on the SPD manifold.

Second, the contributions are not very clear. For example, the derivation of Theorem 3.4 has already been done in eq 17 of "Riemannian Multiclass Logistics Regression for SPD Neural Networks" from Chen et al. Furthermore, it can be directly derived from eq (3) by parametrizing $b_k$ as $\langle p_k, x \rangle$ and then interpreting the subtraction as a Riemannian logarithm.
The distance $d(S, \tilde{H}_{A, P})$ is defined twice: in eq (8) and eq (11). One should be a proposition and the other a definition.
There is a mistake in $b_k$ in Appendix C.

Third, the section 4 is really hard to understand. Specifically, the first paragraph of sub-section 4.1 discusses metrics that have not been presented so far. An example of how to apply theorem 4.2 and lemma 4.3 to a Riemannian metric should be added to understand their implications better.

Forth, the tables in the experiment section are not very clear. For example, in table 4, the authors mention methods [93, 30] and then [93, 70, 30]. What does it mean? The second row of results utilizes the same methods as the first row?

**Questions:**

1) Can you explain more precisely the contributions of the paper? The one presented in the introduction are too broad.
2) Can you provide an example of how to apply Theorem 4.2 and Lemma 4.3?
3) Can you explain the rows in tables of the numerical experiments?

---

> ### Author Response · Authors · 2023-11-19
> **Response Reviewer ur4M (R4) (1/2)**
>
> We thank $\textcolor{blue}{ur4M}$ （$\textcolor{blue}{R4}$）for the careful review and the suggestive comments. Below, we address the comments in detail. ***Notice that as we have revised our manuscript, references to our paper here always denote its revised version, unless otherwise stated.***
> ***
>
> **1. Results of Riemannian multinomial logistic regression (RMLR) on other manifolds.**
>
> Recalling our RMLR in Eq. (14), three operators are involved: Riemannian metric, logarithm, and parallel transportation (or left translation). For specific manifolds, one only needs to put these three operators into Eq. (14). We further implement our RMLR on $\mathrm{SO}(n)$. Please refer to CQ#2 in the common response.
>
> **2. The distance is defined twice: in Eq (8) and Eq (11).**
>
> The distance in Eq. (11) is different from Eq. (8). The distance in Eq. (11) is the one used in this paper. For better clarity, we have change the $d(\cdot)$ in Eq. (8) as $\tilde{d}(\cdot)$.
>
> **3. It can be directly derived from Eq (3) by parametrizing $b _k$ as $\langle p_k, x \rangle$ and then interpreting the subtraction as a Riemannian logarithm.**
>
> Thanks for your insightful comment. We can indeed directly interpret subtraction as the Riemannian logarithm and obtain the RMLR in Eq. (14). However, this direct re-interpretation is only an intuitive re-interpretation and has little theoretical support, as it involves no margin distance or margin hyperplane. However, the margin distance and hyperplane play an important role in designing RMLR, such as the ones on hyperbolic[1] and spherical manifolds [2].
>
> Our insight is that RMLR in Eq. (14) is derived from the Riemannian margin distance (Def. 3.3) and hyperplane (Eq. (7)). In this way, it is precise to call Eq. (14) as RMLR.
>
> We assume your concerns might come from our original structure of Sec. 3.2. To obtain RMLR, the margin distance in Def. 3.3 should be first solved. In the original submission, we put the results on margin distance in the appendix and directly presented our RMLR in the main paper. We have added a theorem about the solution of margin distance (Thm. 3.4) between Def. 3.3 and RMLR (Thm. 3.5) in the revised paper.
>
> **4. RMLR has already done in [3].**
>
> Our work differs with [3] both in ***theory*** and ***practice***, and further covers all the results in [3].
>
> **Theory:** Although our RMLR (Eq. (14)) has a similar expression with [3, Eq. (18)], the underlying theories are significantly different. All the results in [3], including Eq. (18) in [3], are confined within SPD manifolds under the metric pulled back from the Euclidean space. However, numerous metrics are not pullback metrics from the Euclidean space, such as BWM, AIM on SPD manifolds [4], the invariant metric on matrix Lie groups [5], and the metrics on the Grassmann and Stiefel manifolds [6]. [3] fails to construct MLR under these metrics. In contrast, our RMLR is more general and can deal with non-flat metrics. Throughout the proof of Thms. 3.4- 3.5, our RMLR only requires geodesic connectedness. Diverse metrics or manifolds in machine learning satisfy this property. ***Therefore, our RMLR has broader application scenarios.***
>
> **Pratice:** We cover all the results in [3] and extensively study diverse SPD MLRs under deformed metrics. In detail, [3] only implements SPD MLR under $(\alpha,\beta)$-LEM, while our work implements five families of deformed metrics. Besides, as we discussed in Rmk 4.4, when $\theta=1$, our SPD MLR under $(1,\alpha,\beta)$-LEM covers the SPD MLR in [3].
>
> **5. More precise contributions of the paper.**
>
> This work presents a general framework for building Riemannian classifiers on manifolds and specifically showcases our framework on the deformed SPD manifolds. Please refer to CQ#1 in the common response for a detailed explanation.

---

> > ### Author Response · Authors · 2023-11-19
> > **Response Reviewer ur4M (R4) (2/2)**
> >
> > **6. How to apply Thm. 4.2 and Lem. 4.3.**
> >
> > **Thm. 4.2 helps to tune the hyper-parameter $\theta$**
> >
> > The significance of deformed metrics lies in their interpolation between different metrics. Therefore, understanding the limiting cases could help us to tune the hyper-parameter $\theta$. In detail, one can decide $\theta$ for a specific deformed metric around its deformation boundary. This is how we select $\theta$, as shown in App. G.1.
> >
> > For the specific Thm. 4.2, it describes the limiting cases ($\theta=1$ and $\theta \rightarrow 0$) of the deformed metrics of LEM, EM, and LCM. For example, for LEM, the candidate values of $\theta$ \{0.5, 1, 1.5\} are around the deformation boundaries of $\theta=0$ and $\theta=1$. We would not consider the values far from the deformation boundaries, such as $\theta=5$.
> >
> > **Lem. 4.3 provides a succinct proof of deformed SPD MLRs**
> >
> > Sec. 4 aims to present specific SPD MLRs under five families of deformed metrics. Note that all five deformed metrics are pullback metrics, which can be uniformly processed. Therefore, we present Lem. 4.3 to generally solve the RMLR under deformed metric, circumventing the lengthy ad hoc proofs.
> >
> > Our specific SPD MLRs in Tab. 2 are all derived by Lem. 4.3. In detail, the SPD MLR under $2\theta$-BWM is derived by Eqs. (20-21), while other SPD MLRs are derived by Eqs. (18-19). Please refer to App. F.3 for detailed application of Lem. 4.3 for each metric. Perhaps we did not clarify this relation clearly in the original paper. We have highlighted this relation in our revised paper (P6).
> >
> >
> > **7. The rows in Tabs. 3-4**
> >
> > Thanks for your careful review. To thoroughly validate the effectiveness of our MLR, we conduct experiments under different network architectures on SPDNet. The first column in Tab. 3-4 denotes the network architectures. For example, [93, 70, 50, 30] in Tab. 4 denotes the architecture of 3 BiMap layers (App. B.4). We have highlighted this explanation in P7. For better clarity, we have changed the first column's name from "methods" to "Network architectures".
> >
> > >[1] Ganea O, Bécigneul G, Hofmann T. Hyperbolic neural networks.
> > >
> > >[2] Lebanon G, Lafferty J. Hyperplane margin classifiers on the multinomial manifold.
> > >
> > >
> > >[3] Chen Z, Song Y, Liu G, et al. Riemannian Multiclass Logistics Regression for SPD Neural Networks.
> > >
> > >[4] Thanwerdas Y, Pennec X. O (n)-invariant Riemannian metrics on SPD matrices.
> > >
> > >[5] Iserles A, Munthe-Kaas H Z, Nørsett S P, et al. Lie-group methods.
> > >
> > >[6] Edelman A, Arias T A, Smith S T. The geometry of algorithms with orthogonality constraints.

---

> ### Comment · Reviewer_ur4M · 2023-11-20
>
> I thank the authors for all the answers. I read the new version of the paper and it is now much clearer. I am still a little bit disappointed by the numerical experiments where the effects of the metric could have been shown with figures (heat maps, line plots, ...) instead of tables.
>
> Overall, the paper is original and can bring value to the community of machine learning on Riemannian manifolds. Therefore, I raised my rating to a 6.

---

> ### Author Response · Authors · 2023-11-20
> **Thanks for raising the score and suggestive comments**
>
> We thank $\textcolor{blue}{ur4M}$  ($\textcolor{blue}{R4}$)  for raising the score, and we are delighted that you appreciate the value of our work.😄
>
> **1. Visualization of experimental results.**
>
> Thanks for your suggestive comments. Following your suggestion, we visualize histograms of our experimental results on the HDM05 and Radar datasets. Please check G.4 in the appendix. If our paper gets accepted, we will move some experimental settings into the appendix and put these histograms into the main paper.

---

### Official Review · Reviewer_Qd7L · 2023-10-31

**Soundness:** 2 fair
**Presentation:** 2 fair
**Contribution:** 2 fair
**Rating:** 5
**Confidence:** 4

**Summary:**

This paper studies five families of deformed parameterized Riemannian metrics, developing diverse SPD classifiers respecting these metrics. The proposed methods were examined in radar recognition, human action recognition, and electroencephalography (EEG) classification tasks.

**Strengths:**

The paper studies different metrics for classification on SPD manifolds.  The theoretical discussions provide nice insights covering different metrics extending the current solutions proposed in the literature.

**Weaknesses:**

The proposal on the “unified framework” is an overclaim.  The paper provides nice results and in detailed theoretical discussions for different metrics. However, there are still rooms for exploration to develop a “unified framework” such as extension of the work for SPD manifolds with different structures, transformations and classifiers. Therefore, I suggest authors revising their claim considering the concrete results given in the paper, i.e. employment of 5 additional metrics for classification on SPD manifolds.

Although theoretical discussions on different formulation of the metrics are nice, they should be extended considering their complexity and equivalence properties.  In addition, experimental analyses should be extended with additional datasets and backbones.

**Questions:**

-	Can you provide a comparative analysis of complexity (memory and running time footprints) of different metrics, both theoretically and experimentally (e.g. even for one task)?

-	The accuracy of models are sensitive to hyperparameters of the metrics. How can researcher estimate these hyper-parameters in practice?

---

> ### Author Response · Authors · 2023-11-19
> **Response Reviewer Qd7L (R3) (1/2)**
>
> We thank $\textcolor{green}{Reviwer Qd7L}$ ($\textcolor{green}{R3}$) for the instant reply and thoughtful comments. We respond to the questions point by point as follows. ***As we have revised our manuscript, references to our paper here always denote its revised version.***
> ***
>
> **1.  “unified framework” --> "general framework".**
>
> Thanks for your suggestive comments. Our RMLR in Eq. (14) is a general framework, as it only requires the existence of Riemannian logarithm $\mathrm{Log} _P Q$ for any pair $P, Q$ in the manifold $\mathcal{M}$. We call this property geodesic connectedness, and diverse manifolds in machine learning satisfy this property. Therefore, our framework can be implemented beyond SPD manifolds. We have extended our framework into the Lie group. Please refer to CQ#2 in the common response for more details.
>
> Although most of the existing manifolds in machine learning are geodesic connectedness, we do not know if, in the future, there would be a powerful metric with the unknown Riemannian logarithm. Therefore, as you said, "unified" might be a bit overly absolute. We have changed the word "unified" to "general".
>
> **2. Experimental analyses should be extended with additional datasets and backbones.**
>
> Our RMLR in Eq. (14) can readily apply to other manifolds or metrics. Please refer to CQ#2 in the common response for the experiments on $\mathrm{SO(n)}$.
>
> **3. Comparative analysis of complexity (memory and running time).**
>
> **Memory cost:** Recalling Eq. (14) and Tab. 2, each class $k$ requires an SPD parameter $P _k$ and Euclidean parameter $A _k$ in SPD MLR. Therefore, the memory cost of the SPD MLR under each metric is the same.
>
> **Computational cost:** The key factor of the computational complexity of SPD MLRs under different metrics lies in the number of matrix functions, such as matrix power, logarithm, Lyapunov operator, and Cholesky decomposition. These matrix functions are divided into two categories: one is based on eigendecomposition, and the other is the Cholesky decomposition. The following table summarizes the number of matrix functions in SPD MLR. Therefore, the general efficiency of SPD MLR should be EM>LEM>LCM>AIM>BWM. Note that Cholesky decomposition is more efficient than eigendecomposition. Without deformation, the order should be EM>LCM>LEM>AIM>BWM. We report the average training time per epoch in Tab. 10 in App. G.3. As shown in Tab. 10, EM-, LEM-and LCM-based SPD MLR is more efficient than AIM- and BWM-based MLR.
>
> Table A: Number of matrix functions for each class $k$ in different SPD MLRs. a (b) means the number of matrix functions in the SPD MLR under the deformed (standard) metric.
> | Metric | Eig-based Matrix Functions | Cholesky Decomposition | In Total |
> |:------:|:--------------------------:|:----------------------:|:-----:|
> | LEM    |             1 (1)            |           0 (0)           |  1 (1) |
> | AIM    |             3 (2)            |           0 (0)           |  3 (2) |
> | EM     |             1 (0)            |           0 (0)           |  1 (0) |
> | LCM    |             1 (0)            |           1 (0)           |  2 (1) |
> | BWM    |             5 (4)            |           1 (1)           |  6 (5) |

---

> > ### Author Response · Authors · 2023-11-19
> > **Response Reviewer Qd7L (R3) (2/2)**
> >
> > **4. How can researchers estimate these hyper-parameters in practice?**
> >
> > We have briefly discussed the hyper-parameters in Page 8 and App. G.1. Let us present a more detailed discussion. In fact, for a specific SPD MLR, there are at most three kinds of hyperparameters: Riemannian metric, deformation factor $\theta$, and $(\alpha, \beta)$.
> >
> > The most significant parameter is the choice of Riemannian metric, as all the geometric properties are sourced from a metric. A safe choice would start with AIM or BWM, and then decide whether to explore other metrics further. In detail, as indicated in Tab. 3-6, AIM and BWM generally achieve more robust performance than EM, LCM, and LEM. This could be attributed to the nice property of AIM [1] and BWM [2]. Besides, we also observe the promising performance of MLR under the deformed EM (Power Euclidean Metric (PEM)). Therefore, if efficiency is essential, PEM could be an excellent candidate metric.
> >
> > For the defromation factor $\theta$, one can select its value around the deformation boudary, *i.e.,* \{0.5, 1, 1.5\} for $\theta$ and $\{0.25, 0.5, 0.75\}$ for $2\theta$. The deformation boundary has been systematically presented in Fig. 1. Let us take EM as an example. The $(\theta,\alpha,\beta)$-EM interpolates between $(\alpha,\beta)$-EM ($\theta=1$) and $(\alpha,\beta)$-LEM ($\theta \rightarrow 0$). Therefore, there are roughly three candidate values, $\theta=0.5, 1, 1.5$. Other metrics are similar. We made a more refined selection in our work for an extensive examination. Nevertherless, \{0.5, 1, 1.5\} for $\theta$ and $\{0.25, 0.5, 0.75\}$ for $2\theta$ are enough as candidate values.
> >
> > $(\alpha, \beta)$ can be set as $(1,0)$, as they are less important than the above two hyper-parameters. In detail, as $\alpha$ is less important as a scaling factor [1], one can set $\alpha=1$. $\beta$ only affects the trace part of the Riemannian metric (Eq. (1)), and does not affect Riemannian operators like Riemannian logarithm, exponentiation, and parallel transportation. Therefore, $\beta$ can also be set as 0. In our main paper, to extensively present the effect of each parameter, we also validate the effect of $\beta$. If one wants to do a more refined selection of $\beta$, one can follow the suggestion presented in P8 to tune $\beta$.
> >
> > >[1] Thanwerdas Y, Pennec X. Is affine-invariance well defined on SPD matrices? A principled continuum of metrics.
> > >
> > >[2] Bhatia R, Jain T, Lim Y. On the Bures–Wasserstein distance between positive definite matrices.

---

### Official Review · Reviewer_fSLb · 2023-11-01

**Soundness:** 3 good
**Presentation:** 3 good
**Contribution:** 2 fair
**Rating:** 3
**Confidence:** 4

**Summary:**

The authors present an approach to build classifiers on Riemannian manifolds. This approach is then applied to SPD manifolds with 5 different Riemannian metrics. The proposed method is validated on radar recognition, action recognition, and electroencephalography (EEG) classification.

**Strengths:**

* Summary of notations and mathematical proofs are provided.
* The proposed method improves SPDNet on radar recognition and action recognition, and improves SPDDSMBN on EEG classification.

**Weaknesses:**

* The contribution is incremental as it is heavly based on the works of Nguyen & Yang (2023) and Thanwerdas & Pennec (2019a; 2022a).
* Experimental results are poorly presented (the text size in some tables, e.g. Tabs. 3 et 6 is too small to read).
* Lack of evaluation to show the benefit of the proposed method.
* Limitations are not discussed.

**Questions:**

I have several concerns about the paper (please also see the question):

First of all, there are definitions and statetements that look strange to me.

As stated by the authors, the main motivation behind the proposed approach is that it can be applied to Riemannian manifolds that only require geodesic connectedness as opposed to existing works. However, the definition of geodesic connectedness (Definition 3.1) given in the paper does not seem to be corrected. I'm wondering if "geodesic connectedness" means "there exists a unique geodesic line connecting any two points". As far as I know, the existence of a unique geodesic in Definition 3.1 is too strong. See for instance:

https://www.cis.upenn.edu/~cis6100/diffgeom-n.pdf

It says that a Riemannian manifold is connected iff any two points can be joined by a broken geodesic (a piecewise smooth curve where each curve segment is a geodesic, Proposition 12.10).

Am I wrong ? Please clarify.

This also leads to another question: What are the requirements for the proposed approach to be applicable ? If the requirement is that there must exist a unique geodesic line between any two points of the manifold, then I'm wondering if the range of applicability of the proposed approach is as limited as the approach in Nguyen & Yang (2023) ? Please clarify.

I also doubt the statement at the end of Section 4.2 "our work is the first to apply EM and BWM to establish Riemannian neural networks, opening up new possibilities for utilizing these metrics in machine learning applications". Note that Han et al. (2021) has thoroughly studied the Bures-Wasserstein (BW) geometry for Riemannian optimization on SPD manifolds, where different machine learning applications have been presented.

It is also claimed in the paper that the proposed method is applicable to a broader class of Riemannian manifolds compared to existing works. However, the derived MLRs are all built on SPD manifolds and it is not clear if the proposed method is also effective in improving existing neural networks on other manifolds, e.g. Huang et al. (2017; 2018).

Concerning the experiments, the authors only present comparisons against SPDNet and SPDDSMBN. I could not find any other comparisons against state-of-the-art methods on the target applications in the supplemental material. This makes it hard to make rigorous judgments about the effectiveness of the proposed approach with respect to other categories of neural networks. Taking action recognition application as an example. Many DNNs have been proven effective in this application on large-scale datasets. Experiments on large-scale datasets are thus important to show the advantage of learning on SPD manifolds over other manifolds (e.g. Euclidean, hyperbolic, Stiefel,...).


**Questions:**

In Remark 3.2, it is not clear if item (a) is an observation made by the authors or it is a well-known result in the literature. In the first case, could the authors give a brief proof for that ? Otherwise, the result should be properly cited.


**References**

1. Andi Han, Bamdev Mishra, Pratik Kumar Jawanpuria, Junbin Gao: On Riemannian Optimization over Positive Definite Matrices with the Bures-Wasserstein Geometry. NeurIPS 2021: 8940-8953.

2. Zhiwu Huang, Chengde Wan, Thomas Probst, Luc Van Gool: Deep Learning on Lie Groups for Skeleton-Based Action Recognition. CVPR 2017: 1243-1252.

3. Zhiwu Huang, Jiqing Wu, Luc Van Gool: Building Deep Networks on Grassmann Manifolds. AAAI 2018: 3279-3286

---

> ### Author Response · Authors · 2023-11-19
> **Response Reviewer fSLb (R2) (1/3)**
>
> We thank $\textcolor{brown}{Reviwer fSLb}$ ($\textcolor{brown}{R2}$) for the constructive suggestions and insightful comments! In the following, we respond to the concerns in detail. ***Notice that as we have revised our manuscript, references to our paper here always denote its revised version.***
>
> **1. Our work and contributions are significantly different from Nguyen & Yang (2023) [1] and Thanwerdas & Pennec (2019a; 2022a) [2-3].**
>
> We have briefly discussed the difference in Rmks. 3.6 and 4.4. Here, we present a more detailed explanation.
>
> **Difference with gyro SPD MLR [1]**
>
> Our work differs from gyro SPD MLR [1] both in ***theory*** and ***practice***.
>
> ***Theory:*** Our framework uses no gyro structure and is more general than gyro SPD MLR.
>
> Firstly, the gyro MLR is based on gyro structures. However, not all metrics can induce gyro structures. In contrast, our method only requires Riemannian logarithm and parallel transportation (or group translation), which usually exists in manifolds in machine learning. Our method is thus more general and can be applied to more metrics.
>
> Secondly, even if the gyro structure exists, the margin distance to the hyperplane in Gyro MLR is based on gyro distance and gyro trigonometry [1, Def. 2.22]. Note that gyro distance is defined by the tangent space at the identity [1, Def. 2.15]. However, the most natural counterparts on manifolds are geodesic distance and Riemannian trigonometry (Eq. 12). Therefore, we directly use geodesic distance and Riemannian trigonometry to obtain the margin distance Eq. (13).
>
> Thirdly, Gyro MLR is obtained case-by-case for a specific gyro space, since the pseudo-gyrodistance [1, Def. 2.22] should be specifically solved for each gyro space. On the contrary, our work solves the margin distance in a general form (Thm. 3.4), and presents a general framework in Thm. 3.5. For a specific metric, one only needs to put the Riemannian logarithm and parallel transport into Eqs. (14-15). These operators exist in most of the manifolds in machine learning.
>
> ***Practice:*** Our work implements five families of deformed SPD MLRs (Tab. 2), while gyro SPD MLR only implements gyro SPD MLRs under the standard AIM, LEM, and LCM. Besides, as we discussed in Rmk. 4.4, our SPD MLRs incorporate the results presented in [1]. Moreover, the BWM and EM are geodesically incomplete, while gyro operations require geodesic completeness [1, Eqs. (1-2)]. Therefore, EM and BWM could be problematic to induce gyro structures, as could gyro SPD MLRs. On the contrary, SPD MLRs can be easily obtained by our Thm. 3.5 for these two metrics. The following table briefly summarizes the difference between our SPD MLRs and gyro SPD MLRs. In addition, our RMLR in Thm. 3.5 can also be implemented into other manifolds, such as $\mathrm{SO}(n)$. Please refer to CQ#2 in the common response for our MLR in Lie groups.
>
> Tab. 1 Difference between our SPD MLRs and the gyro SPD MLRs.
> |       SPD MLR      | Metrics Involved | Theoretical Foundations | Geometric Requirement |
> |:------------------:|:----------------:|:-----------------------:|-----------------------|
> |  SPD MLRs  | The standard AIM, LEM, and LCM | Gyro-structures induced by the standard AIM, LEM, and LCM | Gyro-structures, geodesic completeness|
> |  Ours  | $(\theta,\alpha,\beta)$-AIM, $(\theta,\alpha,\beta)$-LEM, $(\theta,\alpha,\beta)$-EM, $\theta$-LCM, and $2\theta$-BWM | Riemannian geometry | Geodesic connectedness|
>
>
>
>
> **Difference with [2-3]**
>
> Although power deformation is discussed in [2-3], they only cover the deformation of $(\alpha,\beta)$-AIM, BWM, and the standard EM. Our paper further generalized $(\alpha,\beta)$-LEM, LCM, and $(\alpha,\beta)$-EM into deformed metrics (see Sec. 4.1). Besides, we not only discuss the properties of all the deformed metrics in Fig. 1 and Tab. 1, but also validate the efficacy of deformation. Experimental results shown in Tabs. 3-6 indicates the effectiveness of the deformed metrics.
>
> **2. Tabs. 3-6 is too small to read**
>
> Thanks for the suggestion. After our work gets accepted, we will divide each table into two lines and move some stuff to supplementary.
>
> **3. Limitations are not discussed.**
>
> The main limitation of our work is the over-parameterized $P _k$ in our RMLR. Recalling our RMLR in Eq. (14), each class would require an SPD parameter $P _k$ and a Euclidean parameter $A _k$. Consequently, as the number of classes grows, the classification layer would become burdened with excessive parameters. We have added a discussion on this limitation in App. A and will address this problem in our future work.

---

> > ### Author Response · Authors · 2023-11-19
> > **Response Reviewer fSLb (R2) (2/3)**
> >
> > **4. Geodesic connectedness and connectedness in Props. 12.10 in [4]**
> >
> > Thanks for your valuable comment. The geodesic connectedness in Def. 3.1 is a bit different from the connectedness in Props. 12.10 [4]. In fact, the connectedness mentioned in Props. 12.10 is a standard term derived from topology. Consequently, Props. 12.10 is presented as a proposition rather than a definition. Intuitively speaking, connectedness characterizes whether the space can be described as a whole. Please refer to [5, Ch. 3.5] for more details on connectedness. In contrast, our geodesic connectedness only describes whether two points can be connected by a geodesic, and we do not think of broken geodesics. Of course, geodesic connectedness implies connectedness, but connectedness does not infer geodesic connectedness.
> >
> > Whether Def. 3.1 is strong depends on the context. For example, on general manifolds, even the geodesic line could be less meaningful as a measurement, as the geodesic distance could be independent of the geodesic line [6, P146]. However, this paper focuses on the Riemannian manifolds in machine learning. In this context, Def. 3.1 can be satisfied by many manifolds and metrics, such as SPD manifolds, Grassmannian manifolds, Stiefel manifolds, and various kinds of Lie groups. To the best of our knowledge, we have not met the metrics that can be used in machine learning but do not satisfy Def. 3.1. We therefore think this requirement is weak in the context of machine learning.
> >
> > Besides, the proof of Thms 3.4-3.5 only requires the existence of Riemannian logarithm $\mathrm{Log} _P Q$ for any pair $P, Q \in \mathcal{M}$. However, we do not find a term in standard Riemannian geometry to characterize this property. As the existence of $\mathrm{Log} _P Q$ literally means geodesic connectedness, we define this property as geodesic connectedness. If you have any better suggestions to characterize this property, we are willing to make changes accordingly.
> >
> > **5. Whether the range of applicability of the proposed approach is as limited as Nguyen & Yang (2023) [1]**
> >
> > Our applicability is broader than gyro MLR. For the gyro MLR, it has two strong requirements: geodesic completeness and the existence of gyro structures. Firstly, the definition of gyro operations [1, Eqs. (1-2)] implicityly require the geodesic completeness. Many incomplete metrics exist, such as BWM, EM, and generalized BWM [8] on SPD manifolds. Secondly, even if a metric is complete, Eqs. (1-2) in [1] should further satisfy the axioms of gyro space [7, Def. 2.3]. On the contrary, our RMLR (Eq. (14)) merely requires the Riemannian logarithm, which is satisfied by various manifolds or metrics in machine learning.
> >
> > Besides, even if a metric admits gyro structure, to obtain gyro MLR, one still needs to obtain the pseudo-gyrodistance to a hyperplane. In contrast, we have finished all the derivations in a general form. Therefore, one only needs to put Riemannian operators in Eq. (14) and then obtain the final formulation. Moreover, as discussed in Rmk. 4.4, our results on the specific SPD manifolds cover the results presented in [1].
> >
> > **6. Doubts on the statement "our work is the first to apply EM and BWM to establish Riemannian neural networks**
> >
> > Although both [9] and our work use BWM, our work differs from [9] in the application of BWM. [9] applies BWM to optimize the SPD parameters in machine learning algorithms. In contrast, we use BWM to directly build Riemannian networks. Constructing optimization algorithms differs from building networks for a specific metric in considering backpropagation (BP). Optimization algorithms do not need to implement BP. However, building Riemannian networks needs to pay special attention to BP, as Riemannian computation usually requires several types of matrix decompositions. For example, in App. F.2.1, we discuss the instability in the BP of parallel transportation of BWM. Besides, we design the numerical stable BP for the Lyapunov operator in BWM in App. F.2.2. Considering the additional efforts to complete the BP and different usages, we therefore think it is safe to claim that we are the first to use BWM for building networks.
> >
> > **7. Application on other manifolds, such as $\mathrm{SO}(n)$**
> >
> > We implement our RMLR on $\mathrm{SO}(n)$. Please refer to CQ#2 in the common response for more details.

---

> > > ### Author Response · Authors · 2023-11-19
> > > **Response Reviewer fSLb (R2) (3/3)**
> > >
> > > **9. Item (a) in Remark 3.2**
> > >
> > > This can be derived by definition. Let us formalize this claim and present the proof. Besides, we have added the following lemma and proof in App. C.
> > >
> > > **Lemma 1.** A manifold $\mathcal{M}$ is geodesically connected iff $\mathrm{Log} _P Q$ exists for any pair of $P, Q \in \mathcal{M}$.
> > >
> > >
> > > *Proof:* $\Leftarrow:$ This can be directly obtained by the definition of Riemannian logarithm. The existence of $\mathrm{Log} _P Q$ means that there is a geodesic $\gamma$ connecting $P$ and $Q$ with an initial velocity of $\mathrm{Log} _P Q \in T _P \mathcal{M}$.
> > >
> > > $\Rightarrow:$ Two points $P,Q \in \mathcal{M}$ are connected by a geodesic $\gamma(t;P,V)$ starting from $P$ with the initial velocity $V \in T _P\mathcal{M}$. In other words, there is a $t _Q$ such that $Q = \gamma(t _Q;P,V)$. By the homogeneity of $\gamma$, we have $Q = \gamma(t _Q;P,V) = \gamma(1;P,t _Q V)$. As the inverse of Riemannian exponential map $\mathrm{Exp} _P (t _Q V) = \gamma(1;P,t _Q V)$, the Riemannian logarithm is $\mathrm{Log} _P (Q)= t _Q V$.
> > >
> > > Therefore, our Def. 3.1 is equivalent to the existence of $\mathrm{Log} _P (Q)$ for any pair of $P, Q \in \mathcal{M}$.
> > >
> > > > [1] Nguyen X S, Yang S. Building Neural Networks on Matrix Manifolds: A Gyrovector Space Approach.
> > > >
> > > > [2] Thanwerdas Y, Pennec X. Exploration of balanced metrics on symmetric positive definite matrices.
> > > >
> > > > [3] Thanwerdas Y, Pennec X. The geometry of mixed-Euclidean metrics on symmetric positive definite matrices.
> > > >
> > > > [4] Gallier J, Quaintance J. Notes on differential geometry and Lie groups.
> > > >
> > > > [5] Armstrong M A. Basic topology.
> > > >
> > > > [6] Do Carmo M P, Flaherty Francis J. Riemannian geometry.
> > > >
> > > > [7] Nguyen X S. The Gyro-Structure of Some Matrix Manifolds.
> > > >
> > > > [8] Han A, Mishra B, Jawanpuria P, et al. Learning with symmetric positive definite matrices via generalized Bures-Wasserstein geometry.
> > > >
> > > > [9] Han A, Mishra B, Jawanpuria P K, et al. On Riemannian optimization over positive definite matrices with the Bures-Wasserstein geometry.
> > > >
> > > > [10] Brooks D, Schwander O, Barbaresco F, et al. Riemannian batch normalization for SPD neural networks.
> > > >
> > > > [11] Barbaresco F, Brooks D, Adnet C. Machine and deep learning for drone radar recognition by micro-doppler and kinematic criteria.
> > > >
> > > > [12] Kobler R, Hirayama J, Zhao Q, et al. SPD domain-specific batch normalization to crack interpretable unsupervised domain adaptation in EEG.

---

> > ### Comment · Reviewer_fSLb · 2023-11-22
> >
> > I thank the authors for their responses but these responses haven't addressed my major concerns (at least my first two questions haven't been answered by the authors).

---

> > > ### Author Response · Authors · 2023-11-22
> > > **Thanks for the instant reply**
> > >
> > > We thank the reviewer for the instant reply. We assume that the two questions unaddressed are the correctness of Def. 3.1 and the applicability of our work. Let us make further clarifications.
> > > ***
> > >
> > > **1. Def. 3.1 v.s. Props. 12.10 in [1].**
> > >
> > > Our Def. 3.1 and Props. 12.10 in [1] are actually two different concepts in general manifolds and topology.
> > >
> > > The ***geodesic connectedness*** in Def. 3.1 is different from the ***connectedness*** in Props. 12.10. The connectedness is a standard term in topology [2, Ch. 3.5], while our Def. 3.1 describes the existence of a geodesic between any pair of data on manifolds. Throughout the paper, we never claim our "geodesic connectedness" is "connectedness".
> > >
> > > Besides, Props. 12.10 presents an iff condition on manifolds for connectedness, while our Def. 3.1 describes the global existence of the logarithm. We present Def. 3.1 because the theoretical results of Thm. 3.4-3.5 relies on the global existence of logarithm.
> > >
> > > In summary, we do not see there are any contradictions between our Def. 3.1 and connectedness in topology or Props. 12.10. The key function of Def. 3.1 is to describe the global existence of the logarithm. Nevertheless, we are willing to change if you have any suggestions for a more proper name for Def. 3.1.
> > >
> > > **2. Applicability of our methods.**
> > >
> > > - Def. 3.1 is not strong in the context of machine learning
> > >
> > > As stated in our earlier rebuttal, whether Def. 3.1 is strong or not depends on the context. For example, As indicated by Props. 12.10 [1], on general manifolds, a pair of data cannot even be connected by broken geodesics, as a manifold could be disconnected. However, we do not deal with these singular cases in pure mathematics. Our context is the manifolds applied in machine learning.
> > >
> > > Many metrics or manifolds in machine learning satisfy our Def. 3.1, such as **the five metrics on SPD manifolds mentioned, Grassmann manifolds, spherical manifolds, Stiefel manifolds, and various matrix Lie groups**. Therefore, our framework can be readily used to develop RMLR on these manifolds or metrics. This is why we can easily showcase the applicability of our RMLR on $\mathrm{SO}(n)$.
> > >
> > > - Our RMLR has broader applicability than gyro MLR
> > >
> > > The gyro SPD MLR requires stronger and more properties than ours. The gyro SPD MLR requires the existence of gyro structures. As we stated earlier, gyro structures require at least geodesically completeness to make gyro operations well-defined [3, Eq. (1-2)]. Apart from completeness, gyro operations should satisfy nine axioms of gyro spaces [4]. These two requirements can not always be satisfied on manifolds in machine learning. For instance, EM and BWM on SPD manifolds are not complete. Besides, apart from the existing gyro structures [3, 5], it is unknown whether other manifolds could induce gyro space, even if their metrics could be complete, such as Stiefl manifolds and various Lie groups.
> > >
> > > In contrast, our RMLR only requires the global existence of the Riemannian logarithm, which is much ***weaker than gyro MLR***. All the mentioned manifolds or metrics above satisfy this condition, including the five metrics on SPD manifolds, Grassmann manifolds, spherical manifolds, Stiefel manifolds, and various matrix Lie groups.
> > >
> > > > [1] Gallier J, Quaintance J. Notes on differential geometry and Lie groups.
> > > >
> > > > [2] Armstrong M A. Basic topology.
> > > >
> > > > [3] Nguyen X S, Yang S. Building Neural Networks on Matrix Manifolds: A Gyrovector Space Approach.
> > > >
> > > > [4] Nguyen X S. A Gyrovector space approach for symmetric positive semi-definite matrix learning.
> > > >
> > > > [5] Ganea O, Bécigneul G, Hofmann T. Hyperbolic neural networks.

---

> > > > ### Comment · Reviewer_fSLb · 2023-11-22
> > > >
> > > > I thank the authors for further clarifications.
> > > >
> > > > However, I respectfully disagree with the authors answers for my questions. The authors said that "...we do not find a term in standard Riemannian geometry to characterize this property...". A quick search shows that the term "geodesic connectedness" has already been used in the literature:
> > > >
> > > > https://arxiv.org/pdf/math/0005039.pdf
> > > >
> > > > https://arxiv.org/pdf/2004.08357.pdf
> > > >
> > > > The definitions of geodesic connectedness in the above works seem to be in line with Proposition 12.10.
> > > >
> > > > So what's the point of giving a new definition for geodesic connectedness ?
> > > >
> > > > The authors are also vague in saying "As the existence of $Log_P(Q)$ literally means geodesic connectedness, we define this property as geodesic connectedness." Does the two properties are the same ?
> > > >
> > > > I believe that the statement "In this context, Def. 3.1 can be satisfied by many manifolds and metrics, such as SPD manifolds, Grassmannian manifolds, Stiefel manifolds, and various kinds of Lie groups." is not true. Grassmann manifolds are counter-examples because of the existance of cut locus points.

---

> ### Author Response · Authors · 2023-11-22
> **Thanks for further clarification and insightful comments.**
>
> Thanks for the instant reply and insightful comments! According to your comments, we have revised our paper to ensure our claims are more precise. We address your concerns in the following.
>
> **1. Geodesic connectedness --> The existence of the Riemannian logarithm**
>
> Thanks for your insightful comments. We carefully read the literature you present. In the first literature [1], geodesic connectedness refers to whether each two points in the manifold can be joined by a geodesic, which is a bit different from our previous definition (our previous definition is defined by a unique geodesic).
>
> Nevertheless, the ultimate reason for presenting this definition is that we want a short name to describe the existence of the Riemannian logarithm. We have changed the "geodesic connectedness" into "the existence of the Riemannian logarithm" throughout the paper for better clarity.
>
> - **Why we require the existence of the Riemannian logarithm.**
>
> Recall the reformulation of MLR Eq. (6-7):
> $$
> p(y=k \mid S) \propto \exp (\operatorname{sign}(\langle \tilde{A} _k, \mathrm{Log} _{P _k}(S) \rangle _{P _k})\|\tilde{A} _k\| _{P _k} \tilde{d} (S, \tilde{H} _{\tilde{A} _k, P _k})),
> $$
> $$
> \tilde{H} _{\tilde{A} _k, P _k} = \{S \in \mathcal{M}: g _{P _k}( \mathrm{Log} _{P _k} S, \tilde{A} _k) = \langle \mathrm{Log} _{P _k} S, \tilde{A} _k \rangle _{P _k}=0\}.
> $$
> The minimal requirement is the existence of Riemannian logarithm $\mathrm{Log} _{P _k}(S)$ for each $k$. Without the existence of the Riemannian logarithm, even the hyperplane is ill-defined. However, previous work further relies on additional properties to realize MLR in manifolds, such as the generalized law of sines and gyro structures. In contrast, throughout the proof of Thm. 3.3-3.4, we only require the existence of the Riemannian logarithm.
>
> **2. Applicability**
>
> As we clarified, the only requirement is the existence of the Riemannian logarithm. Many metrics or manifolds on machine learning indeed satisfy this property, including the five families of metrics discussed in this paper on SPD manifolds, the Lorentz model on hyperbolic manifolds [2, Eq. (11)], spherical manifolds [3, Sec. 3.1.1], and Lie groups [4, Tab. 1].
>
> In our main paper, we have implemented our framework under five families of metrics on SPD manifolds. Besides, we further implement our framework on $\mathrm{SO}(n)$. These applications should justify the applicability of our approach.
>
> **3. The Grassmann**
>
> Thanks for your insightful comments on the cut locus of the Grassmann. On the Grassmann $\mathrm{Gr} _{n,p}$, for $P, Q \in \mathrm{Gr} _{n,p}$, the Riemannian logarithm $\mathrm{Log} _{P} (Q)$ exists for $Q \in \mathrm{Gr} _{n,p} \backslash \mathrm{Cut}_P$ [5, Props. 5.6], where $\mathrm{Cut}_P$ is the cut locus of $P$.
>
> As we mentioned above, if $S$ is in $\mathrm{Cut} _{P _k}$, the hyperplane and MLR do not exist. In contrast, assuming $S$ is not in $\mathrm{Cut} _{P _k}$ for each $k$, our Thm3. 3.3-3.4 still holds. In [6], the author also makes a similar assumption when dealing with the logarithm on the Grassmann [6, Sec. 3.2, P4]. However, how to ensure or transform $S$ outside $\mathrm{Cut} _{P _k}$ is out of the scope of this work. We will explore this possibility in the future.
>
> > [1] https://arxiv.org/pdf/math/0005039.pdf
> >
> >[2] Liu Q, Nickel M, Kiela D. Hyperbolic graph neural networks.
> >
> >[3] Chakraborty R. Manifoldnorm: Extending normalizations on Riemannian manifolds.
> >
> >[4] Boumal N, Absil P A. A discrete regression method on manifolds and its application to data on SO (n).
> >
> >[5] Bendokat T, Zimmermann R, Absil P A. A Grassmann manifold handbook: Basic geometry and computational aspects.
> >
> > [6] Nguyen X S. The Gyro-Structure of Some Matrix Manifolds.

---

### Official Review · Reviewer_tGbz · 2023-11-05

**Soundness:** 3 good
**Presentation:** 2 fair
**Contribution:** 2 fair
**Rating:** 6
**Confidence:** 3

**Summary:**

This paper presents a unified framework for designing Riemannian classifiers for geometric deep
learning.  In this paper, we presented a  framework for designing intrinsic Riemannian classifiers
for matrix manifolds, with a specific focus on SPD manifolds. The paper studies five
families of deformed parameterized Riemannian metrics. Each of them develops an SPD
classifier respecting one of these metrics.

**Strengths:**

- Extensive experiments conducted on widely-used SPD benchmarks demonstrate that our proposed SPD classifiers achieve consistent performance gains, outperforming the previous classifiers by about 10% on human action recognition,
and by 4.46% on electroencephalography (EEG) inter-subject classification.

**Weaknesses:**

- The presentation of the paper doesn't help the reader to understand the main contributions of the paper.

- The novelty is not clear. Using SPD matrices for human action recognition and EEG is not new.

- Using J. Cavazza, A. Zunino, M. San Biagio, and V. Murino, “Kernelized covariance for action recognition,” in Pattern Recognition (ICPR), 2016 23rd International Conference on. IEEE, 2016, pp. 408–413.
 Eman A. Abdel-Ghaffar, Yujin Wu, Mohamed Daoudi, Subject-Dependent Emotion Recognition System Based on Multidimensional Electroencephalographic Signals: A Riemannian Geometry Approach. IEEE Access 10: 14993-15006 (2022)

**Questions:**

The authors should clarify the novelty of the proposed approach and reorganize the paper.

**Details Of Ethics Concerns:**

NAN

---

> ### Author Response · Authors · 2023-11-19
> **Response Reviewer tGbz (R1)**
>
> We thank $\textcolor{red}{Reviwer tGbz}$ ($\textcolor{red}{R1}$) for the valuable comments. Below is our detailed response.
> ***
>
> **1. Novelty and Contributions**
>
> This paper presents a general framework for building Riemannian Multinomial Logistic Regression (RMLR) on manifolds, and specifically showcases five implementations on SPD manifolds. Please refer to CQ#1 in the common response for detailed discussions about our contributions and novelty.
>
> **2. Using J. Cavazza, A. Zunino, M. San Biagio, and V. Murino, “Kernelized covariance for action recognition,” in Pattern Recognition (ICPR), 2016 23rd International Conference on. IEEE, 2016, pp. 408–413. Eman A. Abdel-Ghaffar, Yujin Wu, Mohamed Daoudi, Subject-Dependent Emotion Recognition System Based on Multidimensional Electroencephalographic Signals: A Riemannian Geometry Approach. IEEE Access 10: 14993-15006 (2022)**
>
> We have added these two papers in the reference of our revised version. Since our SPD MLRs are independent of network structures, our MLRs should be able to be applied to classify the SPD features in these two papers without any problems. We will explore this possibility in the future.

---

### Author Response · Authors · 2023-11-19
**Common Response (1/2)**

We thank all the reviewers for their constructive suggestions and valuable feedback 👍.  Below, we address the common questions (CQs). ***Notice that as we have revised our manuscript, references to our paper here always denote its revised version, unless otherwise stated.***
******
**CQ#1: Novelty and contributions ($\textcolor{red}{R1}$ and $\textcolor{blue}{R4}$)**

This paper proposes a general framework for designing Riemannian Multinomial Logistic Regression (RMLR) for Riemannian networks. Specifically, our work has three main contributions:

1. We develop a framework for designing RMLR on general Riemannian manifolds in Sec. 3.2.
2. For the specific SPD manifolds, we systematically generalize five families of popular Riemannian metrics into deformed metrics in Sec. 4.1.
3. To showcase our RMLR framework, we propose five specific SPD MLRs induced by deformed metrics in Sec. 4.2.

Let us explain more about our contributions.

**1. General framework for RMLR**

Our framework of RMLR is more general than the existing methods, as it only requires geodesic connectedness (Def. 3.1). As discussed at the end of Sec 3.1, the RMLRs in [1-3] are confined to specific metrics. Their methods require some special geometric properties, such as the generalized law of sines, Gyro-structures, geodesic completeness, and pullback metrics from Euclidean spaces. However, some metrics or manifolds do not have these properties. For example, Bures-Wasserstein Metric (BWM) does not have these properties. Even if BWM has nice properties for measuring SPD manifolds [4], it is difficult to build SPD MLR based on [1-3].

By contrast, our RMLR (Eq. (14)) only requires geodesic connectedness. In machine learning, most of the metrics or manifolds satisfy this property, including SPD [4], Cholesky manifolds [5], and many kinds of matrix Lie groups [6]. This indicates that our methods have broader applicability. Besides, as discussed in App. C, our RMLR is a natural generalization of Euclidean ones.

**2. Deformed metrics on SPD manifolds**

As shown in [7], the matrix power function can interpolate between different metrics on SPD manifolds. Only AIM, BWM, and standard Euclidean metrics are generalized into deformed metrics in the existing literature. In contrast, we further extend LEM, LCM, and two-parameter EM into deformed metrics and systematically study the deforming effect of these metrics, including their Riemannian properties (Tab. 1) and limiting cases (Thm. 4.2 and Fig. 1). Experimental results shown in Tab. 3-6 demonstrate the deforming effect of these metrics.

**2. Deformed SPD MLR**

For the specific SPD manifolds, we propose five families of deformed SPD MLRs induced by five families of deformed metrics. Although our framework does not use Gyro-structures, our SPD MLRs incorporate the MLRs presented in [2-3]. Besides, none of them develop SPD MLRs under BWM or Euclidean Metric (EM). In contrast, we systematically discuss five deformed families of SPD MLRs. We briefly summarize the difference in the following table. We have added this discussion in App. H.

Tab. 1 Difference between our SPD MLRs and the previous SPD MLRs.
|       SPD MLR      | Metrics Involved | Theoretical Foundations | Geometric Requirement |
|:------------------:|:----------------:|:-----------------------:|-----------------------|
|  [2]  | The standard AIM, LEM, and LCM | Gyro-structures induced by the standard AIM, LEM, and LCM | Gyro-structures, geodesic completeness|
|  [3]  | $(\alpha,\beta)$-LEM| Pullback metric from the Euclidean space |Pullback metric from the Euclidean space |
|  Ours  | $(\theta,\alpha,\beta)$-AIM, $(\theta,\alpha,\beta)$-LEM, $(\theta,\alpha,\beta)$-EM, $\theta$-LCM, $2\theta$-BWM | Riemannian geometry | Geodesic connectedness|

---

> ### Author Response · Authors · 2023-11-19
> **Common Response (2/2)**
>
> **CQ#2: RMLR on $\mathrm{SO}(n)$** ($\textcolor{brown}{R2}$, $\textcolor{green}{R3}$, and $\textcolor{blue}{R4}$)
>
> Our RMLR (Eq. (14)) can readily apply to other manifolds. We focus on LieNet [8], where the latent space is $\mathrm{SO}(3)$. We call RMLR in $\mathrm{SO}(3)$ as LieMLR.
>
> **Theoretical ingredient**
>
> To implement RMLR (Eq. (14)+Eq. (15).), we need the Riemannian metric, Riemannian logarithm, and parallel transportation, which can be found in [9, Tab. 1] and [10, Sec 3.2.1]. We denote $\log$ as the matrix logarithm and skew-symmetric matrices as $\mathfrak{s o}(n)$, which is the Lie algebra of $\mathrm{SO}(n)$. Supposing $S,P \in \mathrm{SO}(n)$, $V _1,V _2, \in T _P \mathrm{SO}(n)$ and $V \in \mathfrak{s o}(n)$, we present all the necessary ingredients in Tab. 1.
>
> Table 1: Riemannian operators required in LieMLR on SO(n)
> |       Operator      |          Expression          |
> |:--------------------------:|:----------------------:|
> |Riemannian metric |$\langle V _1, V _2\rangle _P=\langle V _1, V _2 \rangle$|
> |Riemannian logarithm |$\mathrm{Log} _P(S)=P \log (P^{\top} S)$|
> |Parallel transport from $I$ and $P$|$\Gamma _{I \rightarrow P} (V)= PV$|
>
> According to Thm. 3.5,  the RMLR on $\mathrm{SO}(n)$ is given as
> $$
> p(y=k \mid S ) \propto \exp ( \langle \mathrm{Log} _{P_k} S,  \Gamma _{I \rightarrow P _k} ({A}_k) \rangle _{P_k})= \exp (\langle \log(P ^\top _k S), A _k \rangle)
> $$
> where $k$ denotes $k$-th class, $S \in \mathrm{SO}(n)$ is an input $\mathrm{SO}(n)$ feature, and $P _k \in \mathrm{SO}(n)$ and $A_k \in \mathfrak{so}(n)$
>
> **Implementation details and dataset**
>
> As LieNet was originally implemented by Matlab, we use the open-sourced Pytorch code [11] to reimplement LieNet by torch. We found that the matrix logarithm in the Pytorch code [11] fails to deal with singular cases well. Hence, we use Pytorch3D [12] to calculate the matrix logarithm. For the LieMLR layer, $P _k$ is optimized by the Riemannian SGD on $\mathrm{SO}(3)$, while $A _k$ is optimized by Euclidean SGD.
>
> In the original LieNet, classification is carried out by Euler axis and angle representation. We substitute this layer with our LieMLR. We call LieNet with our LieMLR as LieNetLieMLR. Following LieNet, we adopt the G3D-Gaming dataset [13], and follow all the settings of LieNet-3blocks in the original paper of LieNet. Similar to LieNet, we train the network by the standard cross-entropy loss.
>
> **Results**
>
> The results are presented in the following table. Due to different software, our reimplemented LieNet is slightly worse than the performance reported in [8]. However, we still can observe the improved performance of LieNetLieMLR over LieNet.
>
> Table 2: Results on LieNet with or without LieMLR
> |       Methods      |          Result          |
> |:--------------------------:|:----------------------:|
> |LieNet |86.06|
> |LieNetLieMLR |87.88|
>
>
>
> >[1] Ganea O, Bécigneul G, Hofmann T. Hyperbolic neural networks.
> >
> >[2] Nguyen X S, Yang S. Building Neural Networks on Matrix Manifolds: A Gyrovector Space Approach.
> >
> >[3] Chen Z, Song Y, Liu G, et al. Riemannian Multiclass Logistics Regression for SPD Neural Networks.
> >
> >[4] Thanwerdas Y, Pennec X. O (n)-invariant Riemannian metrics on SPD matrices.
> >
> >[5] Lin Z. Riemannian geometry of symmetric positive definite matrices via Cholesky decomposition.
> >
> >[6] Hartley R, Trumpf J, Dai Y, et al. Rotation averaging[J]. International journal of computer vision.
> >
> >[7] Thanwerdas Y, Pennec X. The geometry of mixed-Euclidean metrics on symmetric positive definite matrices.
> >
> >[8] Huang Z, Wan C, Probst T, et al. Deep learning on lie groups for skeleton-based action recognition.
> >
> >[9] Boumal N, Absil P A. A discrete regression method on manifolds and its application to data on SO (n).
> >
> >[10] Chakraborty R. Manifoldnorm: Extending normalizations on Riemannian manifolds.
> >
> >[11] https://github.com/hjf1997/LieNethttps://github.com/hjf1997/LieNet
> >
> >[12] Ravi N, Reizenstein J, Novotny D, et al. Accelerating 3d deep learning with pytorch3d.
> >
> >[13] Bloom V, Makris D, Argyriou V. G3D: A gaming action dataset and real time action recognition evaluation framework.
>
> ***
> For brevity, we refer to reviewers $\textcolor{red}{tGbz}$ as $\textcolor{red}{R1}$, $\textcolor{brown}{fSLb}$ as $\textcolor{brown}{R2}$, $\textcolor{green}{Qd7L}$ as $\textcolor{green}{R3}$, $\textcolor{blue}{ur4M}$ as $\textcolor{blue}{R4}$, respectively.

---

### Author Response · Authors · 2023-11-22
**Gentle Reminder (24h Left)**

Dear Reviewers,

We sincerely appreciate your dedicated time and effort in reviewing our paper. There are 24 hours left in the reviewer-author discussion period. As some reviewers have not acknowledged reading our response, we are uncertain if all concerns have been adequately addressed. If you have additional questions, we are more than willing to clarify further.

Your feedback is crucial to us, and we look forward to further discussions. 😄😄

Best regards,

Authors

---

### Meta-Review · Area_Chair_7wbJ · 2023-12-07

**Metareview:**

This paper present an approach to build intrinsic classifiers on Riemannian manifolds, with a focus on the manifold of SPD matrices with 5 different Riemannian metrics. In particular, the authors proposed a generic Riemannian Multiclass Logistic Regression (RMLR) algorithm for geometric deep learning. The proposed method is validated on several experiments on real data sets.

Reviewers generally appreciate that the improvements in empirical results of the proposed method over SPDNet. However, they also point out that the paper has limited novelty compared with previous work, including Thanwerdas & Pennec (2019a; 2022a), Nguyen & Yang (2023), and Chen et al (2023a). During the reviewers-authors discussion period, the authors revised the paper and clarified some of the technical issues raised. However, the following key issue remains after this revision:

- Two definitions of *margin distance* are provided, one is simply called *margin distance* (Eq(8)) and the other is called *Riemannian margin distance* (Eq(13)). These two definitions are not consistent with each other, yet the Riemannian margin distance is plugged in Eq(6) in place of the margin distance to obtain the RMLR. This inconsistency must be resolved.

**Justification For Why Not Higher Score:**

The theoretical formulation is mathematically flawed.

**Justification For Why Not Lower Score:**

N/A

---

### Decision · Program_Chairs · 2024-01-16

Reject